# Interdisciplinary strategy to assess the impact of meteorological variables on the biochemical composition of the rain and the dynamics of a small eutrophic lake under rain forcing

Fanny Noirmain[1], Jean-Luc Baray[2], Frédéric Tridon[3], Philippe Cacault[4], Hermine Billard[1], Guillaume Voyard[5], Joël Van Baelen[6], Delphine Latour[1]

*Correspondence to*: Fanny Noirmain (fanny.noirmain@gmail.com)

[1]Université Clermont Auvergne, CNRS, Laboratoire Microorganismes : Genome, Environnement (LMGE), UMR6023, Clermont-Ferrand, France

[2]Université Clermont Auvergne, CNRS, Laboratoire de Météorologie Physique (LaMP), UMR6016, Clermont-Ferrand, France

[3]Department of Environment, Land and Infrastructure Engineering, Politecnico di Torino, Turin, Italy

[4]Université Clermont Auvergne, CNRS, Observatoire de Physique du Globe de Clermont Ferrand (OPGC), UAR833, Clermont-Ferrand, France

[5]Université Clermont Auvergne, CNRS, Institut de Chimie de Clermont Ferrand (ICCF), UMR6296, Clermont Ferrand, France

[6]Université de la Réunion, CNRS, Météo-France, Laboratoire de l'Atmosphère et des Cyclones (LACy), UMR8105, St Denis de la Réunion, France

**Abstract**

We present an interdisciplinary investigation of the links between the macro and microphysical properties of rain, the biochemical composition of rain, meteorological parameters, and their impacts on the phytoplankton dynamics of a mountain lake. In order to document this interdisciplinary scientific question, Lake Aydat in the French Massif Central mountains was fitted with a set of high-resolution atmospheric radars, a disdrometer, and a precipitation collector in 2020. In parallel, the lake was monitored via sensors and water sampling. To illustrate the potential of this novel experimental setup, we present a case study of a rain event that occurred in September 2020 and during which three contrasting sub-periods were identified based on the type of rain (convective or stratiform). Using our high temporal-resolution monitoring, we show that the origin of the air mass mainly influences the chemical composition of the rain, which depends on the rain type. In contrast, the photosynthetic cell concentration in the rain is mainly influenced by meteorological variables, predominantly below-cloud scavenging. The very low concentrations of photosynthetic cells found in rain events cannot directly impact the lake's phytoplankton abundance. In contrast, the rain rate directly impacted the lake's thermal stratification during the convective event. The response of the phytoplankton depends on the genus and interestingly, three cyanobacteria, *Microcystis*, *Coelomoron*, and *Merismopedia*, showed a systematic response to rain events with a sudden decrease in abundance at the lake surface immediately after rain events. In contrast, the abundance of green algae (*Elakatothrix*), picocyanobacteria (*Synechocystis* and *Synechococcus*) and diatoms (*Asterionella and Melosira)* gradually

increased following the rain events, but with a lower intensity compared to the cyanobacteria species. These different phytoplankton responses to the same rainfall event could play a key role in phytoplankton dynamics in the temperate zone. Our results highlight the importance of high-frequency and time resolution monitoring of both atmosphere and lake to better understand the adaptive strategies of cyanobacteria following rain events.

## 1. Introduction

Global warming impacts the Earth's surface temperature, which has increased globally by more than 1°C since the pre-industrial period. In addition to this long-term warming trend, most climatic predictions point to extreme events for many regions, including an increase in frequency and intensity of heavy precipitation in several regions of northern Europe (Stocker and Intergovernmental Panel on Climate Change, 2013). However, the forecast capacity for extreme rain events, such as convective events, is low due to the use of larger-scale numerical weather analyses (Bannister et al., 2020; Brewster, 2003), whereas convective events have high spatial variability and fast vertical development associated with unstable layers of air and turbulence (Pruppacher and Klett, 2010). Hence, a finer box grid length would be more adapted to resolving the convection processes, which is not always compatible with numerical weather prediction models (Bannister et al., 2020). Nevertheless, recent progress in the model simulations has improved the forecast capacity for extreme events and shows, for example, a potential future increase in their frequency in the south of France (Luu et al., 2020; Fumière et al., 2020).

As a receiving environment, lakes will be impacted by this increase in frequency and volume of rain events. Chemical changes to lake water have often been reported in the literature after rain events (Jennings et al., 2012; de Eyto et al., 2016; Stockwell et al., 2020). Such changes could be directly related to wet deposition from the atmosphere, which could affect the dissolved organic carbon (Jennings et al., 2012) and nitrogen concentrations at the lake surface (Brakke, 1977). However, the storm, associated with a high wind speed, seems to have an indirect effect on the nutrients, generating mixing which pulls deeper water up to the lake surface, and leading to the dilution or concentration of nutrients depending on their previous concentrations (de Eyto et al., 2016; Jones et al., 2008; Jennings et al., 2012; Barbiero et al., 1999). These events can also induce intense mixing in the water column and increase the turbidity and turbulence and, in turn, impact the composition of the phytoplankton community (Gray et al., 2019; Stockwell et al., 2020; Huisman et al., 2004; Blottière et al., 2017). The duration of these abiotic changes following rain events varies and can last from days to years, depending on the severity of the meteorological drivers and also site-specific factors (Jennings et al., 2012; Knapp and Milewski, 2020; Stockwell et al., 2020). Such modifications subsequently impact the dynamics of lakes communities. Some studies suggest that lakes communities might experience trophic state shifts towards increased degrees of heterotrophy as a result of extreme rain events (Jennings et al., 2012) prompted by climate change. The response of phytoplankton communities is varied, some showing an increase in abundance of new species well adapted to a mixed environment, others showing a reduction, such as buoyant cyanobacteria (Jöhnk et al., 2008). To date, despite the sanitary issues linked to toxic cyanobacteria, only a few studies have dealt with the effect of rainfall on phytoplankton dynamics in a temperate climate (Wood et al., 2017), with most studies being carried out in subtropical climates, characterized by more extreme rain events (Barbiero et al., 1999; Gaedeke and Sommer, 1986; Jennings et al., 2012; Reynolds, 1980; Stockwell et al., 2020; Znachor et al., 2008). These studies also show

contrasting results after rain events, with either a temporary disruption of cyanobacterial blooms, due to flushing and de-stratification, or an increase or recovery of the cyanobacteria biomass, depending on the time of year, intensity, and frequency of rain events, lake geomorphology, and land use of the catchment area (Reichwaldt and Ghadouani, 2012; Richardson et al., 2019; Rooney et al., 2018; Stockwell et al., 2020; Znachor et al., 2008). Some studies support the theory that abiotic changes to lakes after rainfall events will create favourable conditions for cyanobacterial growth due to a more significant nutrient input after periods of drought, combined with potentially more prolonged periods of high evaporation and stratification (Coumou and Rahmstorf, 2012; Reichwaldt and Ghadouani, 2012).

The rain also acts as a dissemination vector for a number of particles, including photosynthetic microorganisms, which can be present in the atmosphere. Only a few studies have investigated the scavenging of photosynthetic microorganisms by rainwater (Dillon et al., 2020; Wiśniewska et al., 2022) and their diversity and dynamics are still unknown. However, their ability to be washed out from the air column by rain could allow them to colonize new environments. The scavenging of such microorganisms by rain could affect the new ecosystem where they end up (Tesson et al., 2016; Tesson and Šantl-Temkiv, 2018). A recent publication by Dillon et al (2020) estimated a chlorophyll-containing cell flux of between $10^9$ and $10^{12}$ cells per rainy day being incorporated into a lake during a rainy period, which could impact the local water quality and the lake's ecology (Dillon et al., 2020). However, though a link between the chemical composition of rain and the origin of the air mass has been demonstrated (Bertrand et al., 2008), suggesting a long-range transport of inorganic ions, the same processes has not yet been demonstrated for photosynthetic cells in rain.

We present here a method for assessing the impact of rain on the dynamics of lake phytoplankton by characterizing the physical properties of the cloud and rain, the meteorological variables, the origin of the air mass and the biochemical composition of the rain. High temporal resolution data acquisition is needed since these parameters are subject to rapid changes, and this data is then used to study the coupling between lake, precipitation, and atmospheric data. We also compared convective and stratiform rain events to analyse their effects on the scavenging processes and the physico-chemistry of the lake. The instrumental setup, composed of cloud and rain profiling radars, a disdrometer, a precipitation collector, and lake temperature data loggers, was installed at a small eutrophic lake. Additional *in situ* biochemical analyses and photosynthetic cell abundance counts were performed on lake and rain water to complement the atmospheric and lake datasets. This article is structured as follows: Section 2 describes the instrumental setup deployed at Lake Aydat (cloud and rain profiling radars, disdrometer, precipitation collector, and lake temperature data loggers, *in situ* biochemical analysis and photosynthetic cell abundance count in lake and rain water) including the strategy for rain and lake monitoring. Section 3 provides the first results of this instrumental setup, with a case-study from a two-week period in September 2020. We selected three different rain events during this period to illustrate the high temporal resolution atmospheric monitoring strategy, with additional lake sampling carried out before and after each event. Lastly, in Section 4 we discuss the link between the biochemical composition of the rain, the meteorological variables, the origin of the air mass, the type of rain events, and the impact of the rain on the lake's physical conditions and phytoplankton dynamics.

## 2. Materials and methods

### 2.1. Lake site and instrumental setup

Lake Aydat (45.6°N; 2.9°E) is located in the French Massif Central, around 15 km south west of Clermont Ferrand, at 837 m above sea level (Fig. 1 A & B). It is a natural lake that was formed when the Veyre River was dammed by a basaltic lava flow 7500 years ago. This small eutrophic dimictic lake has a total area of 0.6 km², a catchment area of 300 km², and a maximal depth of 15 m, and suffers recurrent cyanobacterial proliferations. Lake Aydat receives 75% of its input from the Veyre River and 25% from lateral supply around the shores and via direct precipitation (Lavrieux et al., 2013).

The instrumental setup comprises the HOBO data loggers (Onset Computer Corp., Pocasset, MA) located in the middle of Lake Aydat which record the temperature every 20 cm from the water surface to a depth of 2.8 m (Fig. 1C). We also used a YSI ProDSS Multiparameter Water Quality Meter instrument (YSI Incorporated., Ohio, USA) to make intermittent *in situ* measurements of the dissolved oxygen and temperature profiles in the middle of the lake.

### 2.2. Atmospheric instrumental setup

The atmospheric instrumental setup is located 420 m from Lake Aydat, at an elevation of approximately 10 m above the lake. (Fig. 1C). It comprises a cloud radar (Mira35c), a rain profiling radar (MRR), a disdrometer (Parsivel²), and a precipitation collector. The atmospheric instruments were installed at Aydat in June 2020, and operational measurements have been collected from September 2020.

The Mira35c and the Micro-Rain-Radar MRR-pro (METEK Meteorologische Messtechnik GmbH Corp., Elmshorn, Germany) are used to provide vertical profiles of equivalent radar reflectivity, spanning heights of 300 m to 15 km and 30 m to 4 km, respectively. The Mira35c is a cloud radar profiler, a Ka-band Doppler polarimetric radar, with a center frequency of 35 GHz. The Micro-Rain-Radar MRR-pro is a rain profiling radar, with a vertically-oriented K band (24.23 GHz) Doppler, Frequency Modulated Continuous Wave (FMCW). Both provide measurements of Doppler spectra from which the reflectivity and the vertical Doppler velocity are calculated. The cloud and rain radars provide an estimation of the rain rate and the liquid water content deduced from the retrieved reflectivity and Doppler velocity profiles. Given their respective wavelengths, the Mira35c has greater sensitivity and can detect tiny cloud droplets, whereas the MRR is only able to detect large raindrops.

The Parsivel² sensor (OTT Messtechnik, Germany) is an optical instrument designed to measure the diameter and fall speed of raindrops. The measurements are made when individual drops intersect a laser beam with a total sampling surface of 54 cm². The diameter of each drop is estimated from the decrease in intensity of the laser beam received by a photoelectric diode, and the fall speed is estimated by the time taken by the drop to cross the beam. Rainfall rates are calculated by integrating the number and size of the drops.

Like the MRR, the Parsivel² provides the drop size distribution (DSD), i.e., the concentration of raindrops per unit volume per unit size interval (N, in m$^{-3}$ mm$^{-1}$). However, the vertical DSD profiles for the MRR cover heights of 30 m to 4 km, while the observations of the Parsivel² are made at 3 m from ground level.

160

The precipitation collector (EIGENBRODT GmbH & Co. KG, Baurat-Wiese-Straße, Königsmoor), used to collect the rain, is an automated wet-deposition sampler Eigenbrodt NSA 181/S. It is equipped with a sensor to open the system when precipitation occurs, preventing contamination by dry deposition between precipitation events. When the sensor detects rain, the funnel door opens and collects rainwater over a 500 cm² area, and water volume is calculated by a tipping bucket system that counts the number of tips. The carousel holds 16 collection bottles, allowing for sequential sampling of the rainwater. We specified parameters to define a rain event: first, the minimum amount of rain to start a new acquisition was 0.15 mm (i.e., three tips) to open the funnel; second, sample bottles were replaced once the maximum quantity of rain per bottle was exceeded (i.e., 250 ml/bottle); and lastly, the end point of the event was set at 15 minutes after the rain stopped, leading to a bottle change and avoiding contamination between rain events. Rain events were characterised individually, with a discrimination time of 15 minutes, as a previous study had shown significant variability in the chemical composition of rain related to air mass variability and rapid changes in atmospheric concentrations due to aerosol scavenging. (Jaffrezo et al., 1990).

The precipitation rate can be calculated by dividing the collected sample volume by the collection area (500 cm²). The printed circuit board (PCB) reader also gives the sampling time and duration.

In addition, the air temperature (°C) and wind speed (m.s$^{-1}$) are recorded continuously by the Meteo France weather station located at Saint-Genès-Champanelle, around 7 km north east of Aydat.

180

A                                                          B

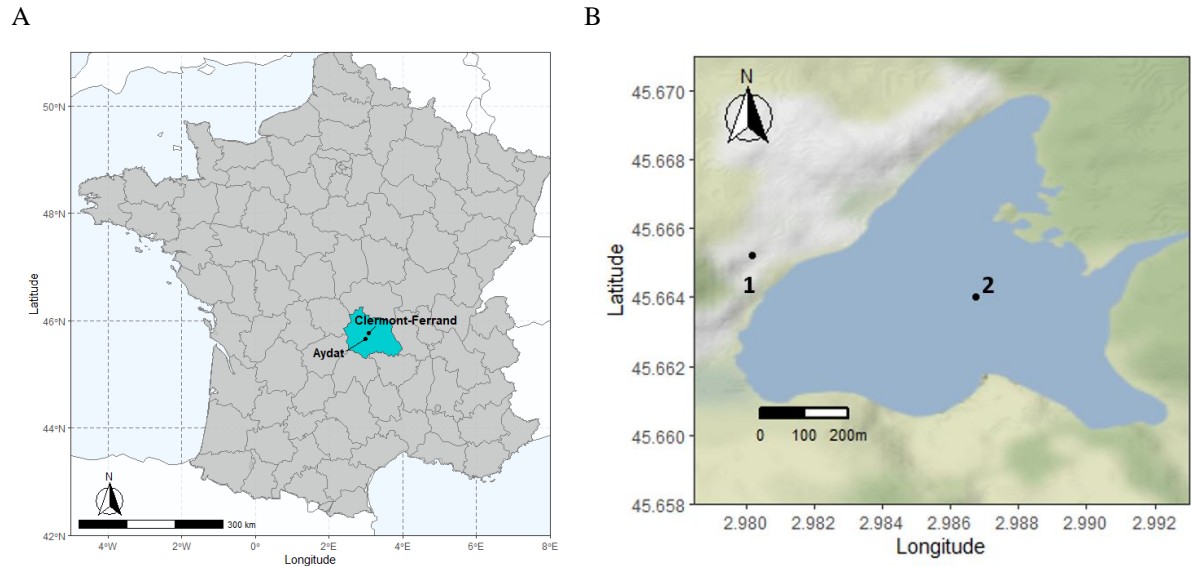

C

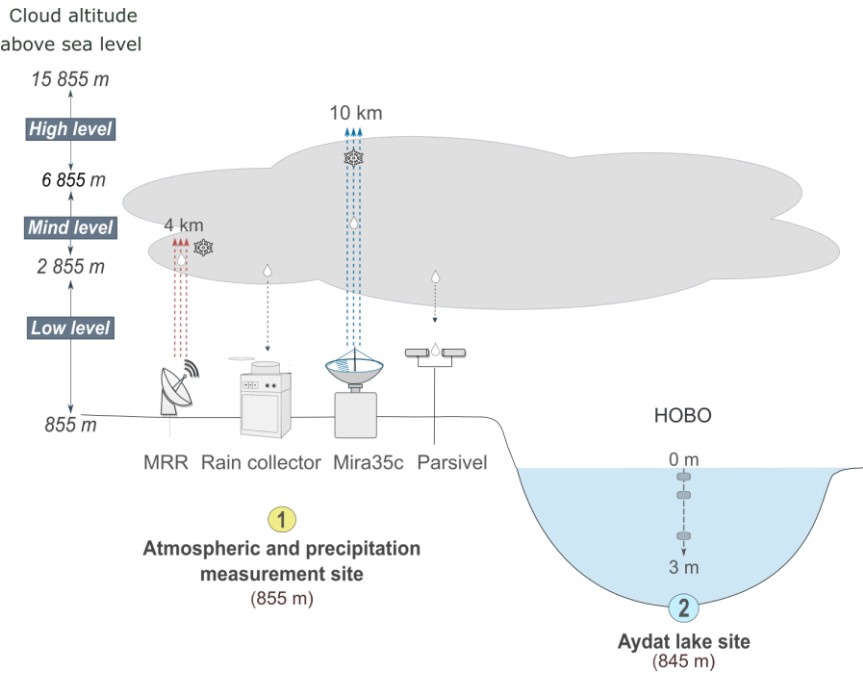

**Fig. 1***:* (A) Location of the town of Aydat, a commune in the Puy-de-Dôme department in Auvergne-Rhône-Alpes in central France. (B) Map of Lake Aydat showing the location of the instrumental setup which comprises (1) the atmospheric and precipitation measurement site, and (2) the Lake Aydat site. (C) Conceptual diagram of the overall Aydat instrumental setup. OpenStreetMap data is available under the Open Database license (https://opendatacommons.org/licenses/odbl/). Maps are released under the Creative Commons Attribution-ShareAlike 2.0 (CC-BY-SA 2.0) license (https://creativecommons.org/licenses/by-sa/2.0/).

### 2.3.  Types of meteorological situations

The rain events studied were classified according to the meteorological nature of the cloud mass as either stratiform or convective, and the rain properties were based on the macro and microphysical characteristics of the cloud and rain events (drop diameter, peak rain rate and mean intensity, and cloud reflectivity). The convective event had higher values for all these physical characteristics and showed a greater degree of variability (standard deviation) due to the air mass being unstable. In contrast, stratiform events are characterized by a more stable air mass, and showed a homogenous cloud reflectivity and low physical variability.

### 2.4.  Back trajectory measurements

We also investigated the air mass history of each rain event using the CAT model. The CAT model is a three-dimensional (3D) forward/backward kinematic trajectory code using initialization wind fields from the recent reanalysis ECMWF ERA-5, with a bilinear interpolation for horizontal wind fields and time and a log-linear interpolation for vertical wind fields. The CAT Model was first used to establish the footprint of atmospheric composition measurements, performed at the Puy de Dôme high-altitude observatory (Baray et al., 2020), and then to classify the cloud sampled, in synergy with the chemical measurements (Renard et al., 2020).

For these measurements, ECMWF ERA-5 wind fields were extracted every three hours with a spatial resolution of 0.5° in terms of latitude and longitude, at 29 vertical pressure levels between 50 and 1000 hPa. Back trajectories to different altitudes corresponding to cloud layers observed on Mira35 radar profiles were calculated, with a temporal resolution of 15 minutes and a total duration of 120 hours. In addition to back trajectory plots, the geographic origin was quantified by counting the number of trajectory points in each of the following nine sectors: north-northeast (NNE), east-northeast (NEE), east-southeast (SEE), south-southeast (SSE), south-southwest (SSW), west-southwest (SWW), west-northwest (NWW) and north-northwest (NNW), plus one nearby area. Points located over land and sea are also counted, by separating those which are close to the ground from those which are in the free troposphere, up to a limit of 2 km above the ground.

### 2.5. Rain and lake monitoring

Monitoring of both the rainfall and the water in Lake Aydat started on September 18, 2020. Since then, all rain has been collected using the precipitation collector and analyzed. The rain water sub-samples were collected in separate bottles to allow sequential sampling. The lake samples were collected before and after each rain period.

We focus on two rain periods of particular interest in this paper to illustrate the potential of the instrumental setup: the first from September 19-21, 2020, named "Rain Period 1" and the second from September 24-28, 2020, named "Rain Period 2" (Fig.2). Within these rain periods, we selected three separate rain events to illustrate our strategy of high-resolution atmospheric monitoring. First, we selected one rain event that occurred on September 20, 2020 from 14:00 to 16:00 UTC, named "High-Intensity-Short-Rain (HIR)" because of the high rain rate observed over a short period of time. Then, from the second rain period, we selected the longest event of this period, named "Continuous rain event 1 (CR1)" which occurred on September 27-28 from 03:50 on September 27 to 01:28 UTC on September 28, and lastly "Continuous rain event 2 (CR2)", which occurred 20 minutes after CR1 from 01:50 to 09:00 on September 28. We considered the CR1 and CR2 as different rain events since the intervening drought period exceeded 15 min.

As well as the high-resolution lake monitoring (HOBO loggers), we carried out additional lake sampling on September 18, 21, and on September 23 and 30 in the middle of the lake (Fig. 2). We also recorded vertical oxygen and temperature profiles using the YSI probe. The lake water was collected between 08:00 and 10:00 UTC at three depths, at the surface, and depths of 1.5 and 3 m, using a Van Dorn horizontal Bottle Water Sampler (2.2L, PVC) deployed vertically with a weight to take it to the desired depth. Then water was transferred into a 15 liter Jerrican to keep the water temperature stable during the transport time back to the laboratory (~20 min).

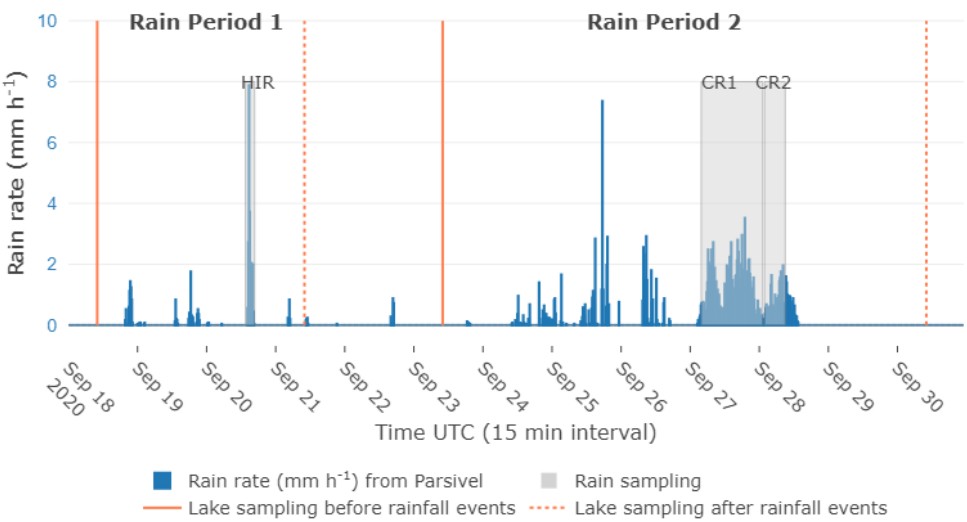

**Fig. 2.** Rain rate intensity from the Parsivel[2] sensor measured at Aydat instrumental site, within a 15 min interval. Lake samplings are indicated by orange lines, the continuous lines indicating sampling performed before rainfall events, and the dotted lines representing sampling performed after rain events. Grey areas indicate the rain sampling periods and their duration, recorded by the Rain sensor. HIR belong to Rain Period 1, and CR1/CR2 belong to Rain Period 2.

### 2.6. Sampling analysis

The rain samples were stored in the dark at 4°C until their analysis within 48 hours of the end of the rain event. We measured the pH of the fresh rain samples and quantified the number of photosynthetic cells by flow cytometry. Then we filtrated the rainwater on a 0.2 μm Nylon membrane, pre-rinsed with 500 ml of Milli-Q water and stored the samples at -20°C until we measured the dissolved nutrients by ion chromatography, which was carried out before the end of 2020.

In parallel, the lake water was immediately filtered in the laboratory using a 150 μm Nylon membrane to avoid the presence of zooplankton in the lake samples. The filtrate (under 150 μm in size) was fixed in a neutral Lugol solution (Sigma-Aldrich) by adding 10 ml of Lugol's iodine stock solution to 150 ml of the filtrated lake sample, after which a counting process was carried out under a microscope following the European Standard NF15204 (AFNOR, 2006). The fixed samples were stored in the dark at 4°C and were counted within the year. For analysis of the major inorganic ions, 1L of the fresh lake sample was filtrated on a 0·2-μm Nylon membrane, pre-rinsed with 500 ml of ultra-pure Milli-Q water to avoid contamination. The lake filtrate was then stored at -20°C until the chromatography analysis, performed within the year. One sample per depth was analysed for each time the measurements were carried out. The mean relative abundance of phytoplankton was normalized using the square root of cell concentration to counter the heterogeneity in the number.

A similar ionic chromatography methodology was used for the rain and lake samples. The dissolved nutrients, $Ca^{2+}$, $K^+$, $Mg^{2+}$, $Na^+$, $NH_4^+$, $Cl^-$, $SO_4^{2-}$, and $NO_3^-$, in the rain and lake water were measured by ion chromatography using a DIONEX ICS6000 chromatograph. Ion chromatography (IC) analysis was performed by

260 a Dionex ICS-6000 equipped with an IonPac AG11-HC (guard-column, $2 \times 50$ mm) and an IonPac AS11-HC (analytical column, $2 \times 250$ mm) for anions and with an IonPac CG-16 (guard-column $2 \times 50$ mm) and an Ion-Pac CS16 (analytical column $2 \times 250$ mm) for cations. The elution conditions are gradient mode of KOH (1 mM to 60 mM in 35 minutes, flow rate 0.36 ml/min) for anions and isocratic mode of MSA (MethaneSulfonic acid at 30 mM, flow rate 0.25 ml/min). Chromatograms were recorded with a conductimetric cell detector using

265 Chromeleon 7.2 software (Thermo Scientific). The nutrient concentrations were measured in triplicate.

We quantified the number of photosynthetic cells in fresh rain samples based on their pigment type using a flow cytometry methodology covering a size range from 0 to 30 μm. Flow cytometry is well adapted to species present in rainwater, which are commonly under 30 μm in size in the atmosphere. However, this methodology is not

270 suitable for quantifying phytoplankton, as they often exceed 30 μm in size, especially the colonial or filamentous forms, leading to an underestimation of their cellular concentration (data not shown). We used conventional bandpass filters to detect the autofluorescence signals from photosynthetic pigments excited by the 640 nm and 488 nm lasers and collected the emissions using FL3, FL2, and FL4 detectors (670 Long Pass,585/42, and 661/16 Band Pass filters respectively) (Dashkova et al., 2016).

275 After each rain event, a blank was systematically performed with sterile water, following cleaning of the rain sensor with detergent, alcohol, and sterile water. Within two days of collection, rain and blank samples were concentrated 100-fold by ultra-fast-filtration (Vivaspin® 100 kDa). First, the membranes fitted to Vivaspin® concentrators were systematically pre-rinsed with 15 ml of Milli-Q water. Then the rain sample was concentrated twice by repeated centrifugation with 15 ml of rainwater in the same Vivaspin® concentrator for 25 min at room

280 temperature (3000 x g). Finally, the concentrate was diluted in sterile Tris-EDTA (TE) buffer before counting the photosynthetic cells by flow cytometry (BD FACSCalibur™).

We used different emission sources and excitation detector channels to distinguish specific pigment populations. The photosynthetic cells were quantified using an air-cooled argon-ion laser, which excited the cells at 488-nm and collected the fluorescence emissions with a FL3 detector (670 Long Pass), a proxy for chlorophyll-containing

285 cell populations, which does not discriminate between the other photosynthetic pigments and bioaerosols (i.e., pollen) emitting fluorescence in this range. In order to distinguish the photosynthetic pigments, the phycocyanin-rich cells were identified out and quantified using a ≈635-nm red diode with an FL4 detector (661/16 Band Pass), whereas the phycoerythrin-rich cells were analyzed using a 488-nm laser in association with the FL2 (585/42 Band Pass) detector (Suppl. Fig.1).The analytical steps consisted of firstly creating a bi-parametric cytogram by

290 selecting FL3 vs. SSC (side scatter) channels to set the "total pigment" population and exclude unwanted debris. Then, from the previous "total pigment" population, we created a second cytogram using FL3 vs. FL2 channels to set the "phycoerythrin" population, allowing us to detect the pink to yellow cyanobacteria and the Cryptophyceae (Read et al., 2014). The last plot was then created from the inverted "phycoerythrin" gate by selecting FL3 vs. FL4 channels to set two independent and specific gates for the "chlorophyll" population, which can detect the

295 Chlorophyceae, Diatoms, and Chrysophyceae; and the "phycocyanin" population, allowing us the detection of blue-green cyanobacteria (Suppl. Fig.1). All gates were systematically calibrated using appropriate phytoplankton species.

The acquisition was performed in high flow rate mode for 60 seconds with detection thresholds applied to FL3 and FL4. Voltage adjustments were carried out to exclude unwanted debris and to optimise placement of the populations of interest.

The exact flow rate was determined using Milli-Q water for each rain sample through the acquisition time and the gravimetric loss from the water sample. Data were acquired in BD CellQuest Pro software and analyzed with BD FACSDiva 9 software (Becton, Dickinson).

### 2.7.    Statistical analysis

The Kruskal Wallis test was performed to determine if the nutrient concentrations in the rainwater varied between the rain events and the CR1 sub-samples, divided into three sub-periods based on changes in reflectivity (Z) detected by the Mira35c. The same analysis was performed between the dates of lake sampling to compare the difference in lake nutrient concentrations before and after a rain period. The Dunn test with Bonferroni correction was applied when the Kruskal Wallis test showed a statistical difference.

The Spearman's rank correlation test evaluated correlations between the lake temperature, recorded with the HOBO sensors, and the rainfall rate, recorded with the Parsivel² sensor (15 min interval), and the wind speed, recorded at Saint-Genes Champanelle (Meteo France, at 1-hour intervals) during the rain events.

A multiple-factor analysis was run using the abiotic measurements from the lake (water temperature, water irradiance, and rain amount), the phytoplankton taxa, and the inorganic ion concentrations in order to assess any correlations between the lake's biotic and abiotic factors during the rain events.

### 3.   Results

### 3.1.    Macro, microphysical and biochemical characterization of cloud and rain events

The three rain events considered showed different atmospheric characteristics (Suppl. Table 1). The "High-Intensity-Short-Rain (HIR)" was a shorter event characterized by high rain intensity, a low wind speed and an air mass of oceanic origin coming from the south-west Atlantic (SWW), traveling mostly at altitudes of greater than 2 km over land and sea (Fig.3 A, Suppl. Fig.2 A & B, Supp. Table. 1). In contrast CR1, occurring during rain period 2, was the longest rain event with a high wind speed marked by an air mass traveling at altitudes of less than 2 km over the land and the brackish water, originating mainly from the continental northeast sectors, with secondary trajectory points close to the Baltic Sea (NNE & NEE sectors) (Fig.4 A-C, Suppl. Fig.2 C-E). CR2, which succeeded CR1 after a 20-min interval without rain, was characterized by similar meteorological parameters to CR1 (i.e., rain rate and wind speed), but with different origins for the air mass: a high percentage of the trajectory points had both continental and brackish water origins, close to the Baltic Sea (NNE) and from the southeast sectors close to the Mediterranean and the Black Seas (SSE & SEE) (Fig. 4 D, Suppl. Fig.2 F, Supp. Table.1 F).

These three events also had different microphysical properties, which were particularly contrasting between HIR and CR1/CR2. HIR can be assigned to a convective type, as the drop diameter, rainfall rate (peak and mean intensity) and cloud reflectivity were all higher and showed greater variability during the event lifetime (standard deviation) (Suppl. Table 1, Fig. 3 B-C). On the other hand, CR1 and CR2 were more similar to a stratiform type due to the lower variability in the drop diameter distribution, rainfall rate, and cloud reflectivity (Suppl. Table 1,

Fig. 4 E-F). In addition, we detected different altitudes for the bright band in the clouds during the three events. The bright band visible in the radar data represents the ice melting layer where the Z factor is higher due to a high reflectivity from the liquid water as it forms around the melting ice crystals (Li and Moisseev, 2020). During HIR, from 14:00 UTC to 15:00 UTC, the bright band was absent, which is often the case during a convective event. In CR1, the rain began to fall from a fairly low height, around 500 m, whereas for CR2 the bright band of rain formation can be seen at about 2000 m, which is further indication of a different air mass for this second rain event (Fig. 3 C 4 F).

The chemical composition of the rain (Fig. 5.A) can also be used to distinguish these rain events, despite the pH remaining quite similar in all the rain samples, between 5.79 and 5.69 (Table 1). The concentration of certain major inorganic ions, $K^+$, $NH_4^+$, $NO_3^-$, and $SO_4^{2-}$ was significantly higher during HIR (p-value adjusted=0.022, 0.023, 0.014, 0.009, respectively) than CR1. In particular, the concentration of $SO_4^{2-}$ was extremely high during HIR, three times more concentrated than for CR1, reaching almost 17 mg L$^{-1}$. Moreover, although CR1 and CR2 had similar microphysical characteristics, the inorganic ion concentrations of CR1 differed significantly from those of CR2. The $Cl^-$, $K^+$, $Mg^{2+}$, and $Na^+$ concentrations were significantly higher for CR2 than CR1 (p-value adjusted=0.014, 0.0048, 0.014, 0.0066, respectively) whereas the concentrations of $Ca^{2+}$ and $PO_4^{3-}$ significantly decreased during CR2 (p-value=0.0073, p-value adjusted=0.022).

Despite a generally low level of photosynthetic cells in rain samples, we reported a high inter-variability between rain events (Fig. 5. B). CR1 was characterized by the highest amount of chlorophyll (3092 cells. L$^{-1}$) and phycocyanin-rich cells (17 cells. L$^{-1}$), whereas HIR was characterized by a lower amount of chlorophyll-containing cells (1126 cells. L$^{-1}$) and the absence of phycocyanin. However, a high intra-variability was observed during CR1, with the phycocyanin pigment detected in only two CR1 samples during the beginning and middle of the CR1 event (Fig. 6C). Phycoerythrin pigment was absent in all three rain events.

### 3.2.    Sequential analysis of CR1 sub-samples

The evolution of the reflectivity (Z) revealed by the Mira35c makes it possible to divide the CR1 event into three sub-periods: firstly, CR1a, characterised by homogenous reflectivity with high cloud above the precipitating zone of the atmosphere (September 27 from 03:50 to 10:30 UTC), then CR1b, with high reflectivity streaks associated with rain bands (September 27 from 10:30 to 20:15 UTC), and finally CR1c, whose low reflectivity is linked to the weakening of the event (September 27 at 20:15 to September 28 at 01:28 UTC) (Fig. 6.A). The mean drop diameter and terminal velocity decreased during the event from CR1a to CR1c (Suppl. Table 2), whereas the highest rainfall rate occurred during CR1b, when the reflectivity was also highest. During this second sub-event, the drop diameter and drop velocity decreased but was more variable overall. At the end of the event, during CR1c, the rain rate, drop diameter, and terminal velocity decreased, associated with the lower Z reflectivity sub-event. During CR1, the pH in the rain samples also decreased from 5.6 to 5.54. The origin of the air mass was principally influenced by the North sectors for all three sub-events. During CR1a and CR1b, the air mass crossed from a continental origin at lower altitude, with a few trajectory points close to the Baltic Sea (NNE & NEE) at an altitude of less than 2 km (Suppl. Table 1). In contrast, during the last sub-event, CR1c, the trajectory points came from

the Northwest oceanic sector (NWW) and from the continental Northeast sector (NEE), still at low altitudes of less than 2 km over land and sea (Fig.4 A-C, Suppl. Fig 2 C-E, Suppl. Table 2).

The statistical analysis carried out on the CR1 sub-samples showed a significant difference in the nutrient footprint between the three sub-events (Fig. 6.B). The Dunn test analysis indicated that $Cl^-$, $NH_4^+$, $PO_4^{3-}$, and $SO_4^{2-}$ were

significantly higher during CR1a compared to CR1b (p-value adjusted= 0.0005, 0.0005, 0.038 and 0.0014, respectively). Further, $Ca^{2+}$, $K^+$, and $NO_3^-$ increased significantly during the last sub-event compared to CR1.b (p-value=0.0073, p-value adjusted=0.022).

We found a high variability in chlorophyll and phycocyanin-rich cell concentrations between the three different

periods. The abundance of photosynthetic cells containing chlorophyll-a decreased consistently between CR1a and CR1c, starting from 4371 cells. $L^{-1}$ and decreasing to 506 cells. $L^{-1}$ at the end of the event. The abundance of cells containing phycocyanin showed a similar variation, with higher concentrations during CR1a and b, dropping to 0 cells. $L^{-1}$ at the end of the event.


A

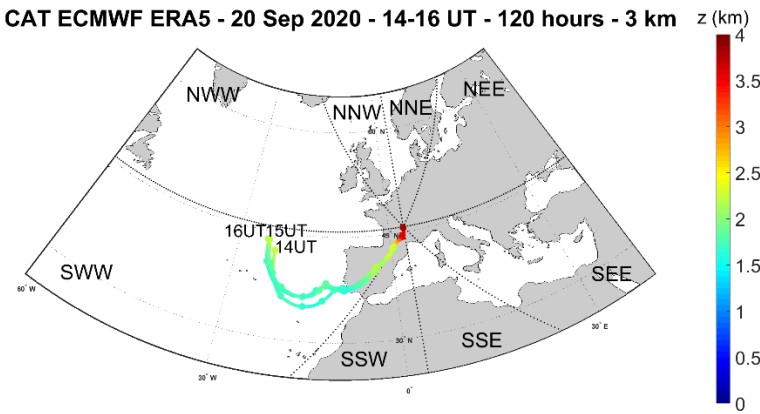

B

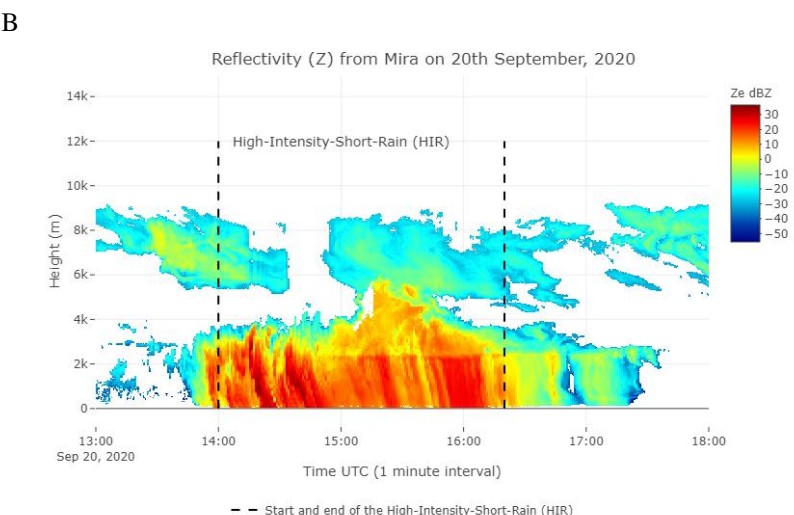

C

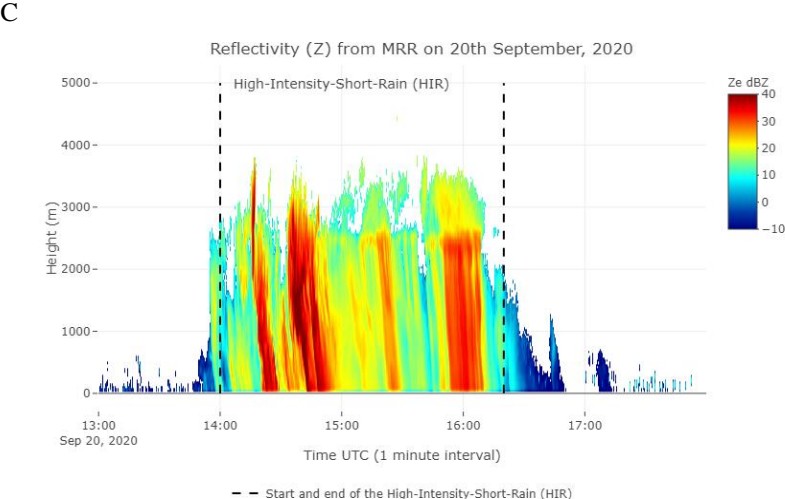

**Fig. 3**: (A) Average of the backward trajectory of 120 clusters calculated every hour, the circles corresponding to the mean calculated every 12 hours during HIR on September 20 from 14:00 to 16:00 UTC at 3 km altitude from the ground level. Time series of vertical profiles of radar equivalent reflectivity from Mira35c on September 20 from 13:00 to 18:00 UTC (B), and with a temporal resolution of 5 seconds from MRR (C).


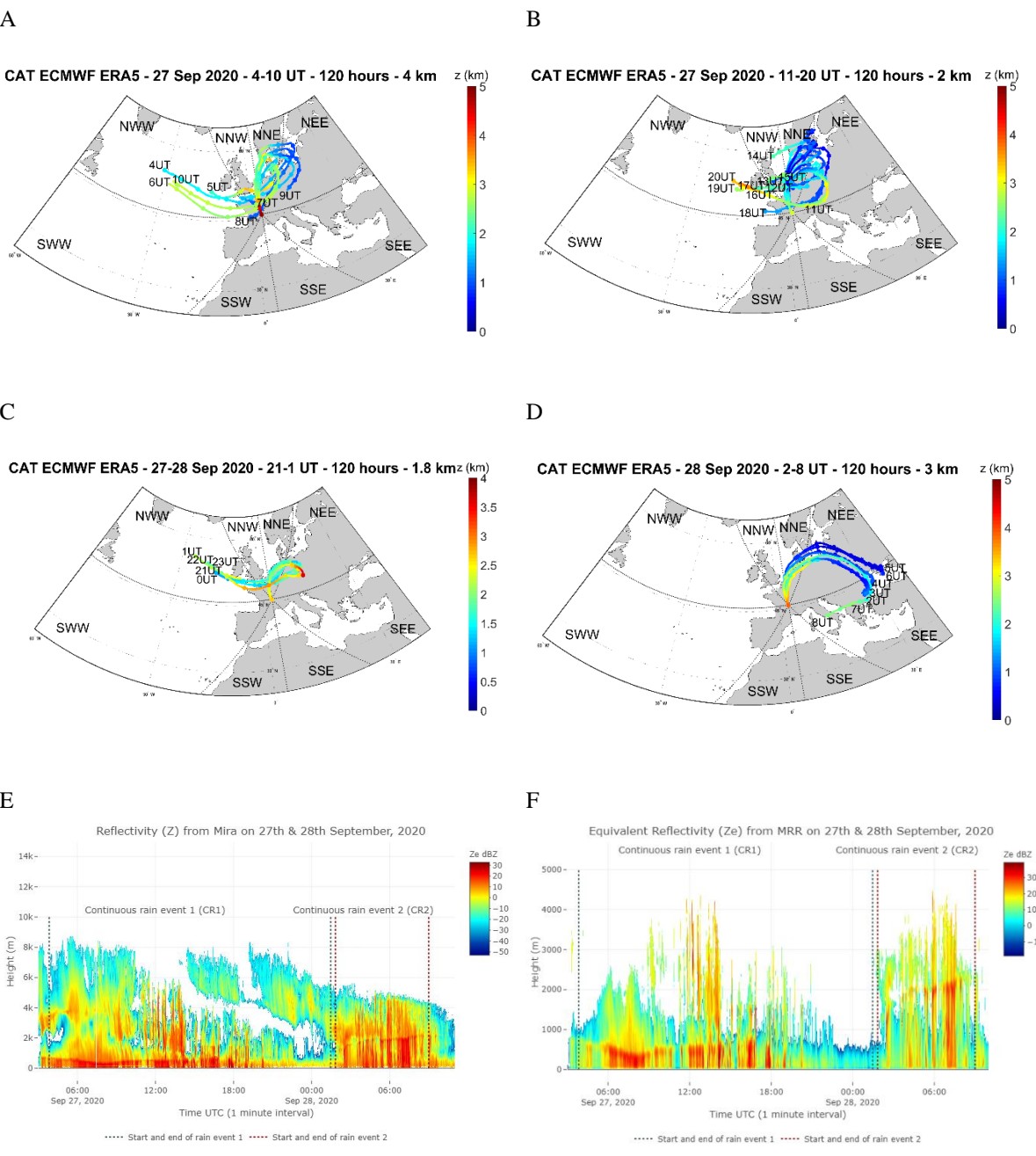

**Fig. 4**: Average of the backward trajectory of 120 clusters calculated every hour, the filled circles corresponding to the mean calculated every 12 hours for (A) CR1a on September 27, 2020 from 04:00 to 10:00 UTC, (B) CR1b from 11:00 to 20:00, (C) CR1c from 21:00 to 1:00 UTC on September 28, and (D) CR2 on September 28 from 1:00 to 9:00 UTC. Time series of vertical profiles of the radar equivalent reflectivity during CR1 and CR2, from September 27 at 03:50 to September 28 at 9:00 UTC from Mira35c (E), and with a time resolution of 5 seconds from the MRR sensor (F).

A

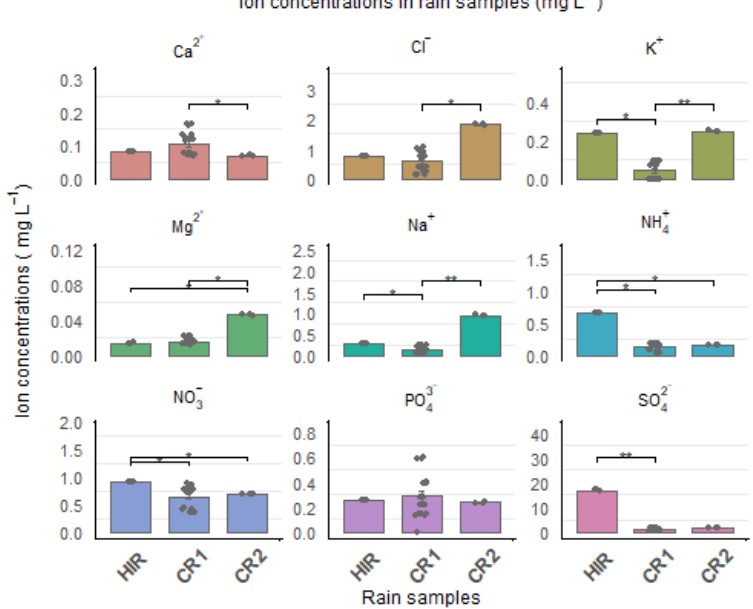

B

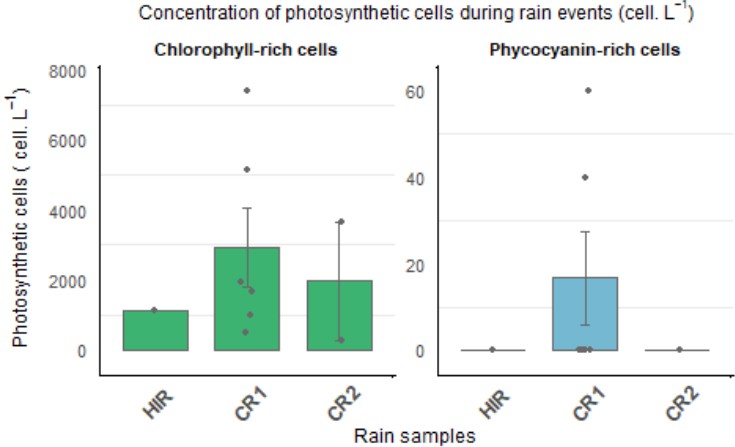

Fig. 5: (A) Inorganic anion and cation concentrations (mg. L$^{-1}$) and (B) Photosynthetic cell concentrations (cell. L$^{-1}$) in HIR, CR1 and CR2. The significance level from the Dunn test is shown to compare the nutrient concentration between rain events.

A

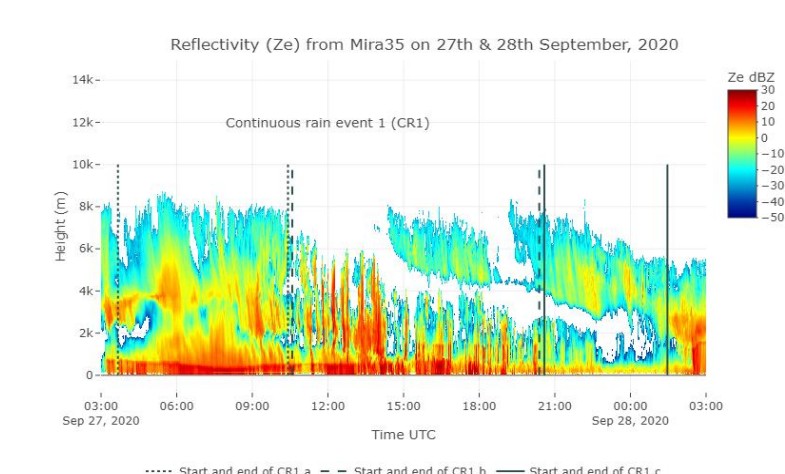

B

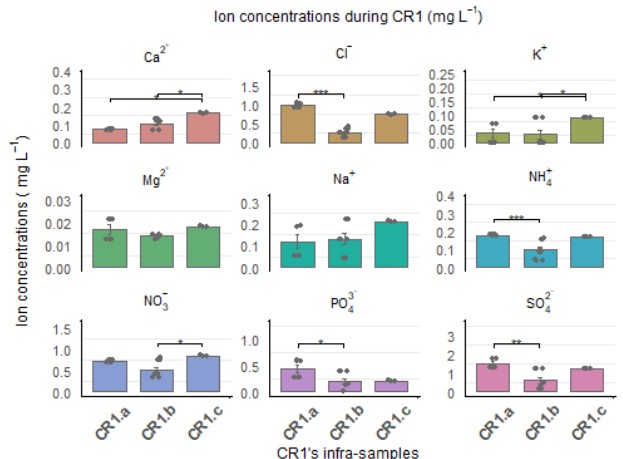

C

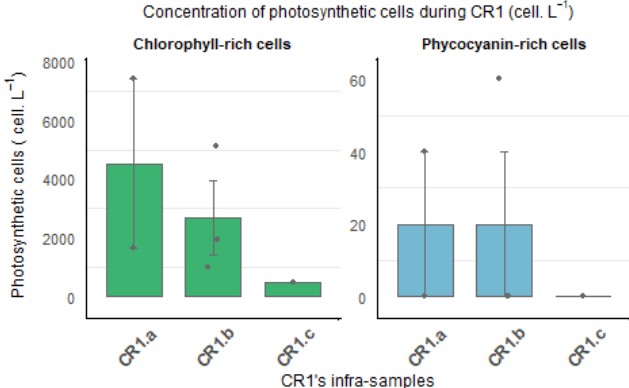

**Fig. 6**: (A) CR1 event divided into three sub-periods based on the evolution of the reflectivity (Z) from Mira35c. (B) Inorganic
nutrient concentrations (mg. L$^{-1}$). (C) Photosynthetic cell concentrations (cell. L$^{-1}$) in CR1 sub-events. The significance level
from the Dunn test is shown to compare the nutrient concentration between the CR1 sub-events.

### 3.3.    Precipitation and meteorological impacts on lake stratification and phytoplankton dynamics

The additional vertical lake temperature profile performed using the YSI probe before and after the rain periods showed that the water responded differently following the two periods (Suppl. Fig.3). After the first rain period, the temperature of the epilimnion (i.e., the upper layer of water in a thermally stratified lake) decreased slightly (1°C). On the contrary, after the second rain period, the epilimnion temperature decreased drastically (6°C), and the thermocline dropped to a depth of 2 m, suggesting a decrease in the vertical stability of the upper water column

(thermocline strength) (Suppl. Fig.3).

The parallel use of the HOBO data provided additional detail of the impact of atmospheric factors on lake temperature structuration during each rain event on a fine scale. For instance, the lake surface temperature before HIR was 0.7°C warmer than at a depth of 2.8m, but this difference was reduced to only 0.26°C following the convective event. Furthermore, HIR was also associated with an air temperature decrease of one degree Celsius,

and with a drastic reduction in the level of water irradiance, from 4032 to 555 W.m$^{-2}$ at the lake surface (Suppl. Fig. 4. A). On the other hand, since CR1 and CR2 began towards the end of the night, when the water surface temperature had not yet been stratified, no secondary thermocline was reported. However, as the level of water irradiance increased to 3400 W.m$^{-2}$ during CR1b, the lake temperature increased slightly at the surface to become 0.4°C warmer than at a depth of 2.8 m. In contrast, a similar temperature was recorded for all depths during CR2,

linked to a low water irradiance level (526W.m$^{-2}$) (Suppl. Fig. 4.B & C).

The relationship between the lake temperature and the rainfall rate varied according to the rain events and the strength of thermal stratification. We found a strong negative relationship between the rain rate and the lake temperature during HIR at the surface (r=-0.97, p=0.0048), which become less significant with depth (Fig. 7 A). In contrast, the relationship between the lake temperature and the rainfall rate during CR1 and CR2 was not

decisive (r> -0.5) and showed a similar trend with depth (Fig.7C&D). The impact of the wind speed was only significant for CR1, for which there was a negative correlation with lake temperature (r=-0.65, p=0.0013) (Fig 7 D).


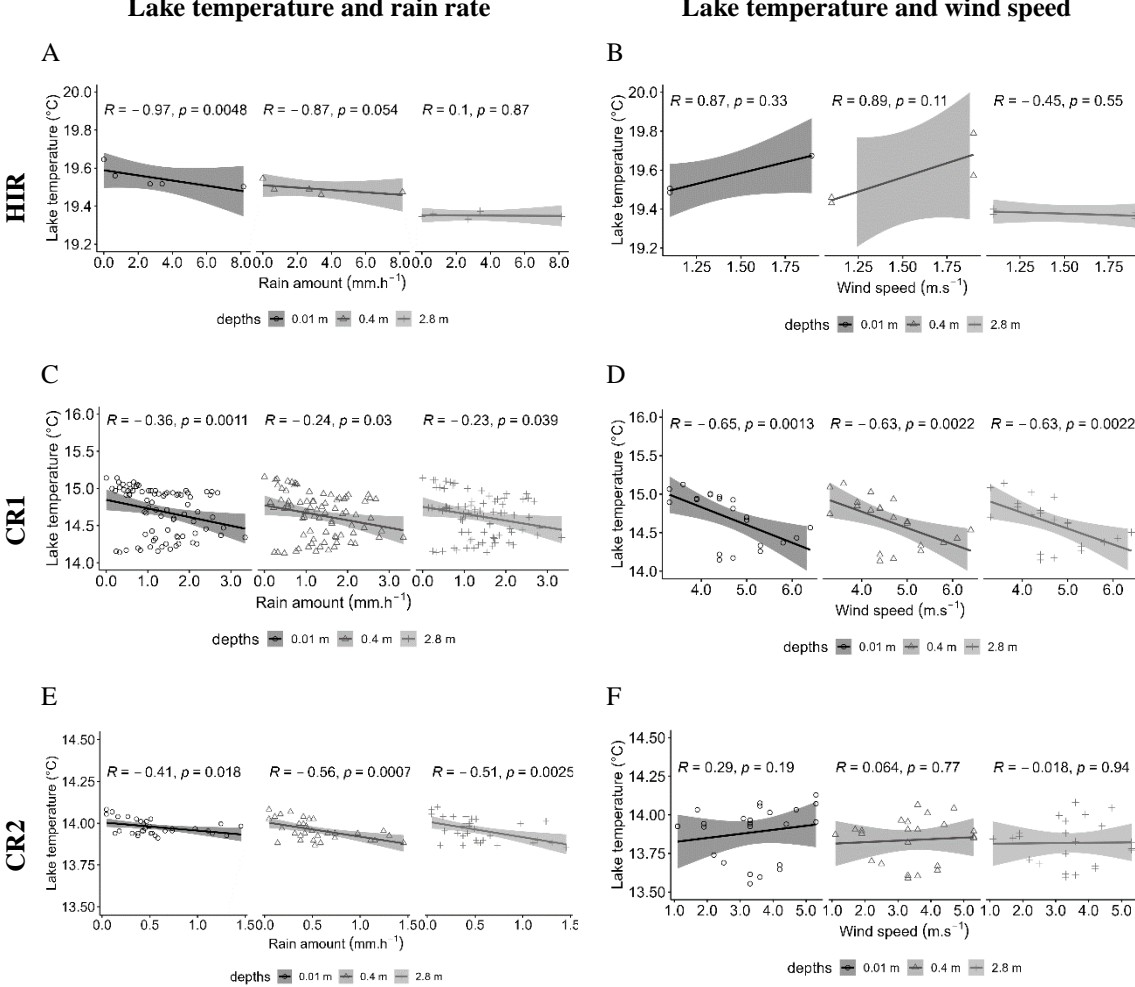

**Fig.7:** Spearman correlations between the lake temperature and the rain rate with the Parsivel (15 min interval) during HIR (A), CR1 (C) CR2 (E), and between the lake temperature and the wind speed recorded by the Meteo France weather station (St-Genes, 1-hour intervals) during HIR (B), CR1 (D) and CR2 (F).


The temporary changes to the vertical thermal structure of the lake during the rain events were accompanied by variations in the concentrations of different phytoplankton species, especially the cyanobacteria. The cyanobacteria concentrations showed high temporal variability (Fig. 8.A) and there was a marked difference in abundance before and after the rain events. *Microcystis*, *Coelomoron*, and *Merispomedia,* decreased systematically

after the two rain periods. For example, *Microcystis* abundance decreased by 60 to 72 % at all depths after HIR, and by 57 to 81 % at all depths after CR. In contrast, despite moderate variations in the abundance of bacillariophytes, charophytes, cryptophytes, and chlorophytes during the lake campaign and between the rain events (Fig8 A), we found a positive relationship between the quantity of rain and the abundance of diatoms (*Asterionella & Melosira*), colonial microalgae (*Elakatothrix*), and unicellular picocyanobacteria (*Synechocystis*)

(Fig 8 B & C). These genera also had a positive relationship with certain ions, such as $SO_4^{2-}$, $Ca^{2+}$ and $NH_4^+$, which increased following the rain events (Fig 8 B & C). The $SO_4^{2-}$ concentration systematically increased after the two rain periods, especially after RP2, by 24 and 32 % at the surface and 1.5 m, respectively (Supp. Fig. 5A). On the

contrary, $NO_3^-$ decreased significantly after the two rain periods (Suppl. Fig. 5A). Of the cations, $Ca^{2+}$ concentration increased systematically after both rain periods (Suppl. Fig. 5B), whereas $NH_4^+$ increased significantly only at the surface after RP1, and at all depths after RP2. No statistical differences were noted between the depth and the abiotic and biotic factors by multiple factor analysis (Fig. 8D).

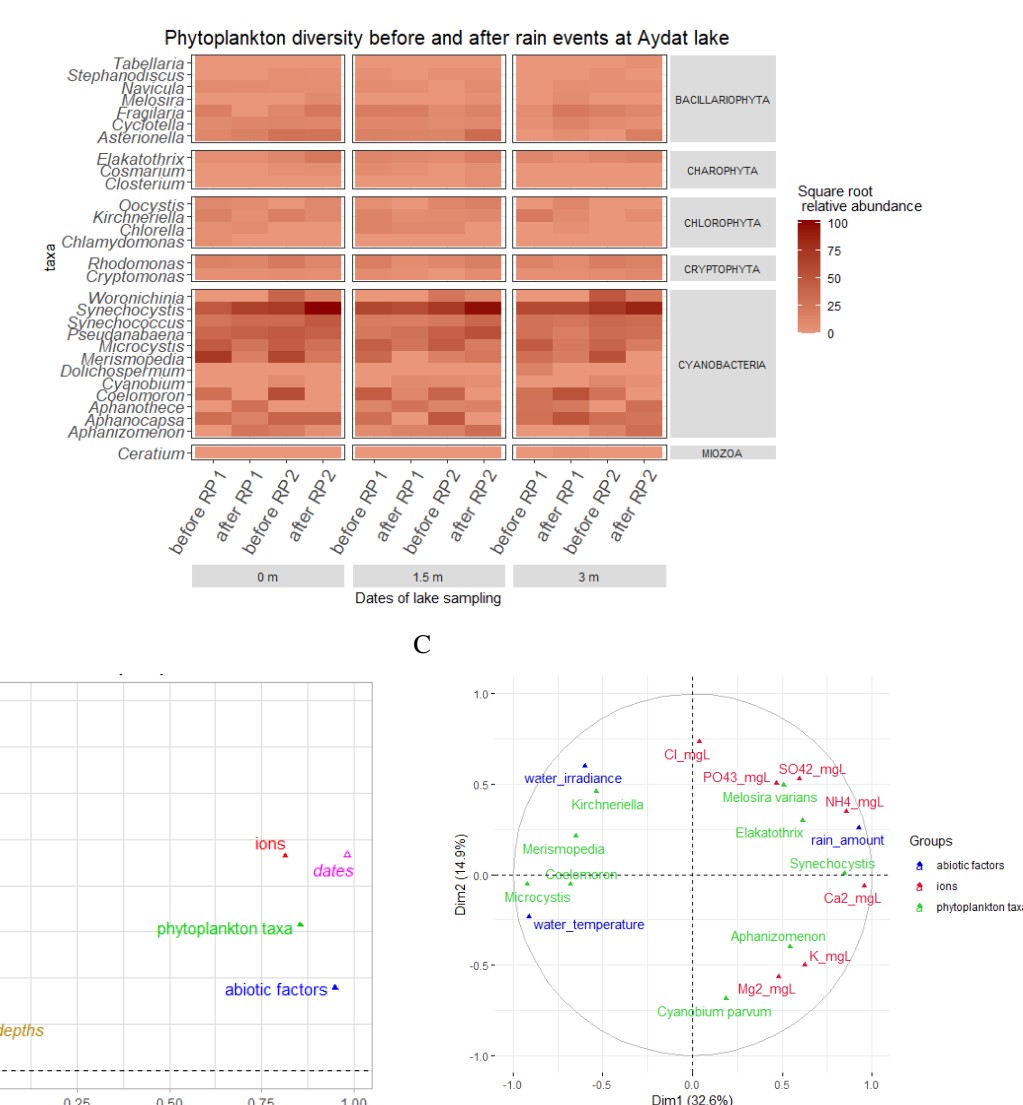

D

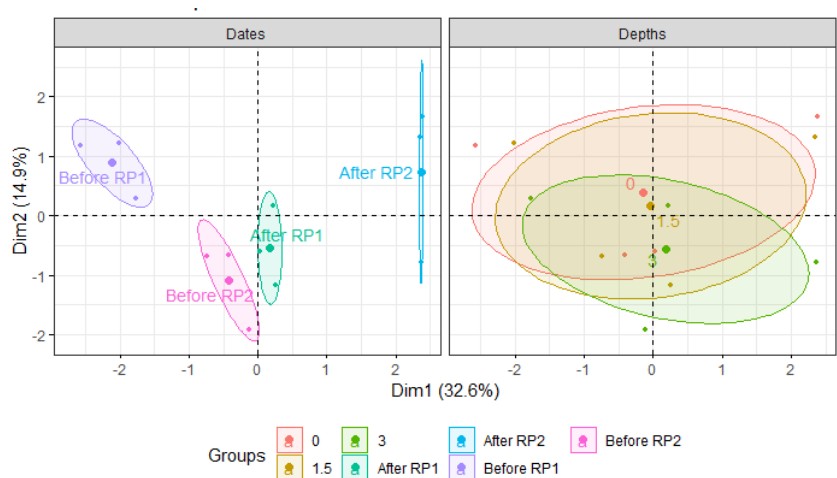

**Fig. 8**: (A) Heatmap showing the square root transformed relative abundance of phytoplankton with respect to depth and the rain periods. (B) Multiple factor analysis (MFA) - projection of groups of variables. (C) Correlation circle map of the MFA. For clarity, only variables with a cos2 higher than 0.45 are represented in the correlation circle. (D) Individual factor map, showing the position of samples in the MFA with confidence ellipses around "depths" (0, 1.5 and 3 m) and "dates" (Before and After Rain Period 1and 2) variables.

## 4. Discussion

### 4.1. Atmospheric processes driving temporal dynamics of inorganic ions and the photosynthetic cell concentrations during rain events.

Our instrumental site enables rain events to be characterized in real-time, and provides a new data set for interpreting the atmospheric processes influencing the biochemical composition of rain. Using the high temporal-resolution monitoring, we confirmed that the chemical rain composition was influenced by long-range transport and local scavenging. Our results provide greater precision on this process, showing that long-range transport could be a major factor influencing the chemical composition of stratiform rain, while local scavenging could be predominant during convective rain. We found a very high concentration of anthropogenic ions during the convective HIR event characterized by an oceanic air mass origin, which is not generally thought to contain a high level of anthropogenic ions (Deguillaume et al., 2014). In addition, based on the $SO_4^{2-}$ to $Na^+$ ratio (Itahashi et al., 2018), chemical analysis of the rain during HIR shows that 99% of $SO_4^{2-}$ came from an anthropic origin. This high concentration is almost twice that measured in previous rainwater studies carried out locally (Opme, located 12 km from our studied site (Bertrand et al., 2008)) and in polluted clouds during a long-term study (Puy de Dôme station, located 6 km from our study site (Deguillaume et al., 2014)). This suggests that below-cloud scavenging was predominant over the cloud mass contribution during our convective event. This could be due to the higher drop diameter in HIR (Suppl. Table 1) which could wash out ions and pollutants more efficiently during a convective event than during a stratiform one. This hypothesis has already been suggested in a previous study that

suggested a drop size of around 2 mm was best for washing out a high number of atmospheric particles (Quérel et al., 2014).

On the other hand, the chemical composition of the stratiform CR events seems to have been highly impacted by the long-range transport, in contrast to the convective event. The CR1 and CR2 events, closer in time, have similar microphysical properties (i.e., mean drop diameter and cloud reflectivity) and occurred during similar meteorological conditions. Nonetheless, they have different biochemical footprints linked to the long-range transport coming from different backward trajectories. Predominantly terrestrial sources marked the CR1 chemical composition, with a few trajectory points from brackish waters (Baltic Sea) (NEE & NNE), agreeing with an east-northeast and north-northeast origin for the air mass, termed "polluted" and "continental" cloud categories. In contrast, the CR2 chemical composition, characterized by continental and stronger brackish origins at low altitudes (close to the Baltic, the Mediterranean and the Black Seas) (NEE & SEE), was mainly associated with $Cl^-$, $K^+$,$Mg^{2+}$,and $Na^+$, confirming the long-range transport of the ions by these air masses, termed "marine" and "continental" cloud categories. The sequential analysis performed with high-time resolution on the CR1 sub-samples confirmed the predominant influence of long-range transport on the chemical composition of rain. In fact, despite meteorological conditions remaining stable, we found an increase in ion reloading at the end of CR1, during CR1c, instead of the expected continual decrease in the ion concentration in these rain samples. These reloaded ions, charged in $K^+$ and $Ca^{2+}$, were associated with the new secondary air mass from the northwest sector, considered as a terrestrial "continental" source.

Long-range transport or local scavenging of the atmosphere could also impact the concentration of photosynthetic cells in the rain. However, information about eukaryotic microalgae and cyanobacteria in rainwater is scarce in the literature, and the scavenging process is not well understood (Dillon et al., 2020; Wiśniewska et al., 2022). Moreover, even if flow cytometry is the most used tool, the methodology used is not always suited to measuring the concentration of chlorophyll-rich cells. If autofluorescence in the FL3 channels is used to detect the concentration of chlorophyll-rich cells, other bioaerosols will also be detected, especially pollen, which has a strong auto-fluorescence in the same wavelength emission range as the photosynthetic cells. Therefore, as the pollen concentration during springtime thunderstorms can be very high and is washed out of the air column by the rain (Hughes et al., 2020), it can artificially increase the concentration of chlorophyll estimate for the rainwater (Negron et al., 2020). Instead, we suggest using the pigment types of different photosynthetic cells to indicate their dynamics, so as not to overestimate concentrations. For this purpose, we developed a method using flow cytometry to separate the concentration of chlorophyll, phycocyanin, and phycoerythrin-rich cells in fresh rain samples (culturing aerial and rainfall species should be avoided as previous studies have indicated problems with culturing aerial bacteria, depending on the main strain involved and the growth medium used (Burrows et al., 2009)). Interestingly, by comparing the scavenging of inorganic ions and photosynthetic cells in the rain, we discovered different dynamics, suggesting that these two parameters are influenced by different atmospheric processes. In contrast with the inorganic ions, the concentration of photosynthetic cells in rainwater was not shown to have a clear link with the long-range transport. Higher levels of chlorophyll-containing cells occurred during CR1, characterized by a predominantly continental origin, while there were lower levels during HIR and CR2, characterized by a more oceanic and brackish influences (Atlantic Ocean and, Baltic, the Mediterranean and the Black Seas, respectively) which are thought to carry a higher load of photosynthetic cells (Wiśniewska et al., 2022). In addition, using the high-temporal resolution during sequential analysis of CR1, we did not see a reloading

of the photosynthetic cells when the new air mass came in from the Northwest sector, which contradicts the hypothesis of Dillon et al. (2020), showing a positive link between the northwest source and the number of chlorophyll-containing cells.

As the major ions necessary to photosynthetic cell growth are present in rainwater, the observed temporal dynamics of these organisms could also be the result of their growth in rain. Nevertheless, due to the very hostile
atmospheric conditions (UV, temperature variation, osmotic shock) (Sialve et al., 2015), this environment is unlikely to support cell growth, although it could be a transitory environment that allows the cells to stay viable before being deposited in a new environment (Després et al., 2012; Amato et al., 2017). As the effect of the cloud mass itself could not explain the temporal dynamics of photosynthetic cells in the rain, we explored the link with below-cloud scavenging and the local meteorological and atmospheric variables.

We found higher photosynthetic cell concentrations during CR1, which was characterized by a higher wind speed, lower air mass altitude, and lower elevation of the bright band in the cloud. In contrast, the lower photosynthetic cell concentrations occurred during HIR, which was characterized by a lower wind speed, higher air mass altitude, and higher bright band elevation. Therefore, these results reinforce the supposition that wind speed is a potential driver for the exchange of aerial microalgae from water reservoirs or other surfaces to the atmosphere through
splash and tap mechanisms (K. Sharma et al., 2006; Rosas et al., 1989; Schlichting, 1964; Tormo et al., 2001). This also suggests that both the bright band elevation and the air mass altitude could influence the microalgae concentration of rain. A similar analysis has already been carried out, suggesting that lower air mass altitude within the planetary boundary layer could be a greater source of biological aerosols due to the long-range movement of air masses (Šantl-Temkiv et al., 2022). As the drop size distribution can influence below-cloud scavenging process,
we studied the link between the microphysical properties of the rain and the abundance of photosynthetic cells in the sequential analysis for CR1. It shows a continual decrease in photosynthetic cell concentration as drop diameter and terminal velocity decrease, suggesting that the washing-out was more efficient at the beginning of the event. However, this decrease could alternatively be due to an impoverishment in the local atmosphere. Because we performed only one sequential analysis during CR1, we would need to acquire more data with high frequency
analysis to confirm this hypothesis of a link between drop size and scavenging potential.

### 4.2.  What are the impacts of rain events on the lake dynamics?

Our results showed that rain events impact the lake temperature, influenced by the rain type and the degree of the
thermal stratification in the lake. We found that during HIR the temperature decreased at the surface of the lake in correlation with the rain rate (r=-0.97, p=0.0048), while the stratiform CR events had no impact on the lake temperature. This suggests that convective events could induce a stronger decrease on the lake temperature than stratiform ones. Further, the higher rain rate and mean drop diameter of HIR probably had a greater impact on the vertical temperature gradient than during the stratiform events, the latter being characterized by a mean drop
diameter of about half the size of those in HIR (Suppl. Table 1) and a lower rain rate. However, as the lake's thermal stratification was not the same during these rain events, it is difficult to isolate individual factors. Wind is also often associated with rain events and has an impact on the lake thermal stratification of a lake. We found the wind's impact to be significant only during the stratiform rain event (CR1) when the higher wind speed (around 5 m.s$^{-1}$) induced a decrease in lake temperature at all depths and weakened the lake's thermal stratification, with the

thermocline sinking by 2 m after CR1 (Supp. Fig.3). However, the onset of autumnal mixing, linked with the decrease in atmospheric temperature during this second rain period probably also contributed to this drop in temperature. The impact of the wind has been described using numerical models (Magee and Wu, 2017; Woolway et al., 2017), showing a notable effect on the lake temperature and stratification by amplifying or mitigating the effect of warmer air temperatures on the thermal structure of the lake, influenced by the direction of the local wind

speed, the fetch, and the geomorphology of the lake. In contrast, the impact of the rain on lake mixing is still not well understood (Stockwell et al., 2020). The studies, mainly carried out in tropical climates, showed that the rain's impact was different and site-specific depending on the lake volume and the runoff volume from the flood (de Eyto et al., 2016). Our work also shows the importance of considering both atmospheric parameters (rain characteristics, wind, air temperature) and lake ones (degree of stratification, seasonal mixing) to better understand

the specific effects of rain on the thermal dynamics of the lake.

We also assessed the impact of the chemical composition of rain on that of the lake to investigate if wet atmospheric deposition could impact the chemical composition of the lake itself. Although some nutrients in the rain were more concentrated than in the lake, we found that the wet atmospheric deposition did not directly affect the biochemical composition of the lake during our short case study. Similar concentrations of nitrogen were

detected in the lake and rain samples, whereas two days after HIR rain, the $NO_3^-$ concentrations decreased at the lake surface down to a depth of 3 m (Fig.5 & Suppl. Fig.5). In contrast, after the first rain period, the concentrations of $Ca^{2+}$, $Na^+$, $K^+$, and $Mg^{2+}$ increased at all depths in the lake, whereas they were between 14 and 190-fold less concentrated in rain water than at the lake surface. Therefore, our results suggest that the wet atmospheric deposition did not influence the ion concentrations in the lake. To our knowledge, no study has compared the ion

concentrations in rain and a lake following rain events. Due to the random occurrence of rain, lake and rain samplings are challenging to perform together. Hence, future works with short measurement periods are recommended to improve our understanding of the causality between the rain's chemistry and the lake's chemical composition, as many ions can be unstable. We also encourage future studies to measure particulate organic matter as some organisms, as is the case for the marine diazotroph Trichodesmium, can promote the dissolution of iron

from dust particles (Rubin et al., 2011).

As all rain events contain photosynthetic cells, we wondered about the role of these organisms on the dynamics of phytoplankton. It has recently been proposed that chlorophyll-containing cell flux enter into lakes, at an estimated rate of between $10^9$ to $10^{12}$ cells per rainy day (Dillon et al., 2020). The authors suggested that this quantity of new

organisms could impact the local water quality and the lake's ecology. However, even if all photosynthetic cells contained in the rain were to be viable, which has not been demonstrated, this assumption did not take into account the ratio between the phytoplankton biomass contained in a lake and the photosynthetic cell abundance in the rain. In general, phytoplankton abundance in freshwater ecosystems ranges from $10^3$ to $10^5$ cells.ml$^{-1}$, as in our study, whereas the concentration of chlorophyll-containing cells in the rainwater is very much lower, reaching a

maximum of 3 cells.ml$^{-1}$ in our case. So, even taking into account the lake surface area, these concentrations are still negligible compared to the total phytoplankton concentration in the overall lake volume. Thus, photosynthetic cells in the rain probably do not impact the phytoplankton biomass. Nevertheless, if these cells were to survive the atmospheric conditions, they could provide a source of new genotypes and enhance the diversity of the phytoplankton of the lake, as previously suggested (Curren and Leong, 2020; Dillon et al., 2020; Wiśniewska et

al., 2019). Our results are based on a unique case study containing three rain events; therefore, future work needs to monitor the short- and long-term diversity to evaluate the consequence of an invasion of new aero-species washed out by the rain in a freshwater ecosystem. Moreover, as we used two different approaches to quantify photosynthetic cells in the rain and the lake due to technical limitations (low amount of rainwater and low abundance of these cells), we encourage researchers to develop a standard methodology assessing diversity in rain

and lake samples.

Although we monitored the lake on a short temporal scale, our results showed that the rain impacted the phytoplankton composition, to different extents depending on the genera, with a particularly contrasting response in terms of the dynamics within the cyanobacteria. There was a systematic decrease in the abundance of certain

cyanobacteria, with a negative correlation between the amount of rain and *Microcystis*, *Merismopedia*, and *Coelomoron* abundances after each rain event. These cyanobacteria belong to Lm and Lo codons and have a low tolerance to mixing (Padisák et al., 2009; Reynolds, 2006; Elliott, 2010; Verspagen et al., 2006). Therefore, their decrease after the rain events could be explained by either an inability to adapt to mixing or by a dilution effect caused by their transport through the entire mixed depth. However, their recovery was relatively quick, as by only

two days after the rain period, the abundances of these three genera were similar to those before the rain at all depths. So, when temporary stratification was restored following the rain, a secondary thermocline nearer the surface helped them to regulate their vertical position by a buoyancy mechanism. It is well known that these cyanobacteria have adaptive strategies for vertical migration to the surface, due to their low surface/volume ratio and gas vesicles (Padisák et al., 2009; Reynolds et al., 2002; Rinke et al., 2010). These adaptive strategies could

help them to recover shortly after a rain event by adjusting their position in the water column (Wood et al., 2017) to benefit from optimal conditions in terms of nutrients and light.

In contrast, other cyanobacteria showed a positive correlation with amount of rain and we found an increase in the abundance of unicellular picocyanobacteria *Synechococcus* and *Synechocystis* after each rain event. In temperate zones, these phytoplankton genera are known to grow in the spring or at the end of summer when the temperatures

are lower and there are enhanced mixing conditions (Callieri, 2010). The negative correlation found between their abundance and water temperature confirms that these species are better adapted to these sub-autumnal environmental conditions. Previous studies have already demonstrated that unicellular picocyanobacteria were able to quickly develop, with an optimal division rate that could double their abundance daily (Callieri, 2010 ; Reynolds, 1994), explaining their rapid increase in our case. In parallel, following the rain events, we also found

a gradual increase in the abundance of some diatoms (*Asterionella* and *Melosira*) and chlorophyceae (*Elakatothrix*) which are also known to be better adapted to lower temperatures and increased mixing conditions (Blottière et al., 2017; Pannard et al., 2007). As the abundance of these species gradually increased during our lake campaign, this trend is probably not especially due to the impact of the rain events but more to seasonal changes associated with the impact of wind and the onset of seasonal mixing.


## 5. Conclusion

The interdisciplinary Aydat instrumental site is designed to carry out high temporal resolution monitoring of the atmosphere and lake, and to improve our understanding of the biochemical and phytoplankton dynamics in a lake following rain events of different magnitudes. Although many lakes are systematically monitored, combined high-frequency observations of the atmosphere and lake are rarely performed simultaneously, and most studies to date have focused on extreme events (de Eyto et al., 2016; Jennings et al., 2012). Our results show that these direct interactions between rain and lake are of particular interest. We found that the origin of the air mass mainly influences the chemical composition of the rain, modified by the rain type, with a greater influence of the long-range transport for stratiform rain, and stronger local scavenging in the case of convective rain. For photosynthetic cells, there was no clear link between their dynamic response and long-range transport of the air mass. Instead, we highlighted a link between the number of chlorophyll-contain cells and the meteorological and atmospheric variables and the below-cloud scavenging. This type of analysis involving the rapid evolution of rain composition is only made possible by high-resolution monitoring. Future studies will be aimed at improving our understanding of the process of how inorganic ions and photosynthetic cells are washed-out from the air column by monitoring a large number of rain samples using high-frequency and time resolution.

Despite our results highlight the non-negligible presence of photosynthetic cells in rain, their very low abundance relative to lake water means that they probably cannot impact phytoplankton dynamics. However, if photosynthetic cells can survive in rain events, they could enhance the diversity of the phytoplankton. Likewise, the wet atmospheric depositions seem not impacted the lake's inorganic nutrients. Nevertheless, our results are based on a unique case study containing three rain events; therefore, future works devoted to the study of both short and long-term monitoring could improve our understanding of the causality between the rain's biochemistry and the lake's biochemical composition.

Using the high-temporal resolution, we showed that rain only impacted the vertical thermal structure of the lake when it was convective and the lake already thermally stratified. The magnitude of this impact depends primarily on meteorological variables, such as solar and water radiation and air temperature, and the physical characteristics of rain events, such as the drop diameter, depending on the type of rain. Phytoplankton communities also responded differently to the contrasting rain events. Interestingly, three species of cyanobacteria showed similar systematic decreases in abundance after rain and recovered shortly a few days after. In contrast, *Synechococcus*, *Synechocystis*, *Elakatothrix* and *Asterionella* increased without interruption following the rain events and during the lake campaign, suggesting that seasonal changes were more influential than the impact of rain. The varied responses of phytoplankton subjected to the same rainfall event are still poorly understood and must be taken into account when seeking to explain the overall dynamics of phytoplankton. Long-term monitoring is necessary to confirm our results. In addition, laboratory experiments and mesocosm studies should be performed to better discriminate the role of the different factors.

**Author contributions**

**Fanny Noirmain**: Writing - Original Draft, Conceptualization, Investigation, Visualization

**Jean-Luc Baray, Joel Van Baelen & Delphine Latour**: Supervision, Review & Editing, Conceptualization

**Frédéric Tridon**: Software-expertise

**Philippe Cacault**: Radar operation and data processing

**Hermine Billard**: Technical support-expertise

**Guillaume Voyard**: Resources-technical support

**Conflict of interest**

The authors declare there is no conflict of interest.

**Acknowledgments**

We thank the FRE (Fédération des Recherches en Environnement), CPER, FEDER, and CAP 20-25, which funded

the atmospheric instruments. We thank "ATHOS environnement" for carrying out the phytoplankton counts in the

lake samples and the members of Team IRTA for their help with the lake sampling. This work was made possible

thanks to the OPGC technical staff (Frédéric Peyrin and Claude Hervier) who managed the atmospheric

instruments installation. We want to thank the Plateforme CYSTEM – UCA PARTNER (Clermont-Ferrand,

FRANCE) for its technical support and expertise and Mathilde Meynier for the optimization of the ion

chromatography methodology. Finally, we thank «Volcans vacances" lodging, for providing us with a place to

install the atmospheric instruments close to Lake Aydat.

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
