# Peer review of "Interdisciplinary strategy to assess the impact of meteorological variables on the biochemical composition of the rain and the dynamics of a small eutrophic lake under rain forcing"

_Biogeosciences, 2022_

## Referee Comment (RC2)

**Interdisciplinary strategy to survey phytoplankton dynamics of $a_n$ eutrophic lake under rain forcing: description of the instrumental set-up and first results**

5 Fanny Noirmain1, Jean-Luc Baray2, Frédéric Tridon3, Philippe Cacault4, Hermine Billard1, Guillaume Voyard5, Joël Van Baelen6, Delphine Latour1

[revised manuscript text omitted]

- А В 100 km Aydat С cloud altitude 15 km 10 km High level 6 km Mind level 4 km 2 km Low level \_\_\_\_\_ ново 0 km MRR Rain collector Mira35c Parsivel 0 m 1
- 160 located at Saint-Genès-Champanelle, around 7 km north east from Aydat.

3 m

2

Atmospheric and precipitation

measurement site (855 m)

(https://opendatacommons.org/licenses/odbl/). Maps are released under the Creative Commons Attribution-ShareAlike 2.0 (CC-BY-SA 2.0) license (https://creativecommons.org/licenses/by-sa/2.0/).

**2.3. Rain and lake monitoring**

170 The monitoring of the rain and Aydat lake started on 18th September 2020. Since then, all rains have been collected with the precipitation collector and analyzed. The infra-samples of rain waters were collected in separated bottles to perform a sequential sampling. The lake samples were collected before and after each rain period.

In this paper, to illustrate the potential of the instrumental setup, we focus on two selected rain periods of particular interest: a first from 19th until 21st September 2020, named "Rain Period 1" and a second from 24th until 28th September 2020, named "Rain Period 2" (Fig.2). Within these rain periods, we have selected three separate rain events to illustrate the strategy of high-resolution atmospheric monitoring. First, we selected one rain event occurring on 20th September 2020 from 14:00 until 16:00 UTC, named "High-Intensity-Short-Rain (HIR)" because of high rain rate observed during a short duration. Then, on the second rain period, we selected the longest

180 event of this period, named "Continuous rain event 1 (CR1)" occurring on 27th September from 03:50 until 01:28. UTC on 28th, and "Continuous rain event 2 (CR2)", occurring 20 minutes after CR1 at 01:50 considered the CR1 and CR2 different rain events as the drought period exceeded 15 min between them. The lake was monitored on 18th and 21st September, and on 23rd and 30th September at three depths from 0.5, and 3 m in the middle of the lake (Fig. 2).

185

190

Fig. 2. Rain rate intensity from the Parsivel2 sensor measured at Aydat instrumental site, within a 15 minutes interval. Lake sampling are indicated by orange line, the continuous orange lines indicate sampling performed before rainfall events, and dotted orange lines represent samplings performed after rain events. All samplings were performed in the morning between 08:00 and 10:00 UTC. Grey areas indicate the rain samplings and their duration, recorded from the Rain sensor. HIR belong to the Rain Period 1 and CR1/CR2 belong to the Rain Period 2.

Lake and rain monitoring

200

230

---

## Author Comment (AC1)

**Comparison of two methods: the flow cytometry and microscopy-based methods for lake and rain samples**

**A) Concentration of photosynthetic cells during rain events (Cell.l⁻¹)**

[Figure]

**B) Concentration of photosynthetic cells during the lake campaign at the three depths (Cell.l⁻¹)**

[Figure]

**C)**
**NOT AVAILABLE**

**D) Microscopy on lake samples (grouped by pigments species type)**

---

## Author Comment (AC2)

A

[Figure]

B

C

[Figure]

**Fig. 9: Multiple factor analysis (MFA). Projection of variable groups (A), Correlation circle map of the MFA, for clarity, only variables with a cos2 equal to and higher than 0.45 are represented in the correlation circle. (B), individual factor map, showing the position of samples in the MFA with confidence ellipses around "depths" and "dates" variables (C).**

[Figure]

**Suppl. figure: Flow cytometry based-method to quantify the autofluorescence from specific pigment from photosynthetic cells in the rain samples. Detection of all aerosols in the rain (A), the total pigments population based on minimal chlorophyll fluorescence (B), Phycoerythrin population (C), Chlorophyll, and Phycocyanin populations (C).**

---

## Author Comment (AC3)

**Comment on bg-2022-100**

**Referee #1**

Referee comment on "Interdisciplinary strategy to survey phytoplankton dynamics of a
eutrophic lake under rain forcing: description of the instrumental set-up and first results"
by Fanny Noirmain et al., Biogeosciences Discuss.,
https://doi.org/10.5194/bg-2022-100-RC1, 2022

We thank referee 1 for his suggestions made online. We respond in blue below to his comments.

The manuscript by Noirmain et al entitled "Interdisciplinary strategy to survey phytoplankton dynamics of a eutrophic lake under rain forcing: description of the instrumental set-up and first results" is well written and easy to read. The described interdisciplinary strategy provides a very useful tool to decipher the impact of atmospheric processes on lake ecology.

An enormous work has been performed on the atmospheric sciences part but the authors only quickly discuss the ecological implications related to their study. The authors also need to consider further factors related the cyanobacteria biology and ecology. This could for instance apply to Lines 400-405 and Figures 8AB.

A negative effect of rain on organism abundance was mentioned for few taxa. Effect can be seen on Fig. 8B on unicellular group forming cyanobacteria (e.g. Woronichinia, Microcystis, Merismopedia...). Whereas some unicellular individual cyanobacteria (e.g. Synechocystis, Synechococcus, Pseudoanabena...) seem to undergo a positive effect. Cell distribution factor is important because it impacts the growth dynamic and survival of the species.

Moreover microalgae can move and migrate through the water column. Almost the same concentration of cells were fund "before" and "after" RP events in the system (Fig. 8A). The increase concentration at a specific depth implies displacement of the phytoplanktonic community or organismal movement (RP2 towards 1.5 m depth, RP1 towards 3 m). Apparently deeper when the rainfall was more virulent (HIR). This is not described in the results nor correlated to Fig. 8B (diversity) but quickly mentioned in the discussion. Was the water column structure more stable (nutrient, temperature, light, water agitation...) at 1.5 m depth in RP2 and at 3 m in RP1?

The water column does not seem more stable at 1.5 or 3 m after the rain events. No change was found regarding water temperature, conductivity, and oxygen concentrations which were similar up to 3 m deep after RP1 (oxygen concentration around 8.72 mg. L-1, conductivity = 116.8 – 117 µS.cm$^{-1}$) and after RP2 (oxygen concentration between 77 and 7.9 mg. L-1 and conductivity between 101.2 and 102.48 µS.cm$^{-1}$). However, as an abiotic factor, only the water irradiance was different and could explain a slightly different distribution of the phytoplankton abundance at 1.5 m after RP2 and 3 m after RP1. Regarding fig 8B, we did not discuss a potential vertical displacement according to the rain type, convective or stratiform, as other rain events have occurred during RP1 and RP2. We analyzed only three rain events to illustrate the monitoring strategy in this article. However, in a future study, more results and observations will help investigate the potential effect of convective or stratiform rain events on vertical lake stability.

All the cyanobacteria taxa retrieved have different ecological requirements. In the discussion Lm and Lo codons are rapidly mentioned. A principal component analysis can further help to characterize which cyanobacteria taxa was favored under certain rain events (e.g. peculiar nutrient signature).

As suggested by the referee, we performed a supplementary analysis by Multiple factor analysis (MFA) (Fig 9, below) using the abiotic lake factors (water temperature and irradiance and rain amount), phytoplankton taxa (described in Fig 8B), and the lake chemical composition (Fig 8C & D) to characterize which cyanobacteria taxa could be favored after rain events. Multiple factor analysis (MFA) is an extension of principal component analysis (PCA) tailored to handle multiple data tables that measure sets of variables collected on the same observations. Such analysis was more adapted considering our set of data. The results are presented in the pdf attached.

The MFA analysis confirms a negative correlation between the rain amount and *Microcystis*, *Merismopedia*, and *Coelomoron* abundances, also confirming the significant change of abundance between the "before" and "after" rain period reported for these species in the results (line 403). Nevertheless, as suggested by the referee, we can add this model to the results (if the ninth figure is authorized in the manuscript) as it also shows a positive correlation between the rain amount and the abundance of diatom (*Asterionella*), colonial microalgae (*Elakatothrix*), and unicellular picocyanobacteria (*Synechocystis*) abundances associated with the increase of some inorganic ions ($SO_4^{2-}$, $Ca^{2+}$ and $NH_4^+$) and the rain amount. In any case, we will indicate the species that increase following the rain amount in reference to Fig 8B.

[Figure]

Fig. 9: Multiple factor analysis (MFA). Projection of variable groups (A), Correlation circle map of the MFA, for clarity, only variables with a cos2 equal to 0.45 are represented in the correlation circle. (B), individual factor map, showing the position of samples in the MFA with confidence ellipses around "depths" and "dates" variables (C).

Moreover, the differences in phytoplankton concentration and taxa diversity between "before" and "after" calls for emission/deposition fluxes (e.g. Dillon et al 2020 doi:10.1128/AEM.01850-20, Mayol et al 2014 doi: 10.3389/fmicb.2014.00557), very briefly mentioned in discussion. However this is of ecological importance. Retrieved cyanobacteria species from the investigated lakes have been reported from atmospheric samples (e.g. Sharma et al 2007 DOI: 10.1111/j.1529-8817.2007.00373.x). Also microalgae can be emitted after local disturbances such as rain (Tesson et al 2016 doi:10.1128/AEM.03333-15, WiÅ›niewska et al 2019, 2022)

from station Fig 1B2 and be redeposited in local system such as station Fig 1B1 or the lake, therefore it is possible that the retrieved peak of Cr1b with highest phycocyanin value could be partly indigenous.

To ensure a correlation between emission and deposition fluxes in Aydat lake, we should sample airborne microalgae close to Aydat lake's surface. Unfortunately, we did not measure the emission fluxes from the Aydat lake and cannot directly link the phycocyanin detection in the rain (from one rain event, CR1) and the increase of picocyanobacteria abundance at the end of lake sampling. Nonetheless, we discuss deposition fluxes by comparing the level of photosynthetic cells collected in the rainwater with those in the lake, suggesting a very low amount of photosynthetic species introduced in the lake by the wet deposition. Moreover, we also discuss the deposition flux that can impact the diversity in the lake by introducing different species collected by the rain (lines 560-567).

I believe that certain of these aspects would enrich the discussion of the manuscript.

Moreover, it can be useful to run the analysis considering the ecology and behavior of these organisms to avoid a Simpson paradox.

The multiple factor analysis was performed, showing the same results as reported in the manuscript.

Minor comments:

Figure 1 - Please add the geographical coordinates on Fig. 1A and the cardinal directions on both Fig. 1A and Fig. 1B.

Supplement Fig. 1 - The geographical coordinates are too small to be readable.

Lines 111 and 113 - Inform about the material manufacturer and country between brackets.

Line 121 - Please add in the text the difference of elevation between the location where the instrumental setup was installed and the lake surface.

Lines 142 and 156 - Spell out the acronyms DSD and PCB.

We will modify the plots, add the material manufacturer and country and spell out the acronyms.

Lines 182-184 - How was the lake sampling performed? Which parameters were investigated? Do the authors refer to Lines 114-116 in situ measurements or were lake water- and/or phytoplankton samples collected? Please describe further.

To provide more details about the lake sampling, we will add this sentence: *"In addition to the lake monitoring, we collected lake sampling at three depths, surface, 1.5 and 3 m deep, using a Van Dorn horizontal Bottle Water Sampler (2.2L, PVC) deployed vertically with a weight to take it to the desired depth. Then water was transferred in 15 Liter Jerrican to keep the water temperature stable during transport back to the laboratory (~20 min). "*

Line 198 - Why was a 10 µm pore size used? In these filtrates, all microorganisms of a size inferior to 10 µm would be present (including. bacteria, cyanobacteria and other <10 µm eukaryotic microalgae), thus affecting the measurements. Moreover, -20°C storage was applied to the filtrates, conditions under which organismal cell damaging occur, releasing further nutrients in the water. Please explain further.

Line 200 - Which Lugol's iodine solution was used (acidic vs neutral)? What was the final concentration used? How and how long before investigation were the samples stored (temperature, darkness, time)? These are important information for cross studies comparison.

We thank the referee for pointing out an inconsistency with the pore size. The correct size of pores used will be corrected in the new version. The correct sentence is that we filtrated the lake water on a 150 μm Nylon membrane to avoid the presence of zooplankton in lake samples. The filtrate (under 150 μm) was fixed in a neutral Lugol solution (Sigma-Aldrich), and 10 ml of Lugol's iodine stock solution was added to 150 ml of lake filtrated samples for the microscopy. For the nutrient analysis, the lake water was filtrated on 0·2-μm pore-size filters Nylon membrane by rinsing the filtrate previously with 500 ml of ultra-pure Milli-Q water to avoid contamination.

The fixed samples using Lugol were stored in the dark at 4°C and were counted within the year. The lake samples filtrated under 0.2μm were kept at -20°C until the chromatography analysis was performed within the year.

Lines 221-236 - How does the flow cytometer detection method differ from existing protocols? (e.g. Haynes et al 2020, https://www.agilent.com/cs/library/applications/application-analysis-aquatic-plankton-novocyte-5994-2112en-agilent.pdf)

We use conventional flow cytometry measurements with bandpass filters to detect the autofluorescence signals from photosynthetic pigments excited by the 640 nm and 488 nm lasers and collected using FL3, FL2, and FL4 detectors (670 Long Pass,585/42, and 661/16 Band Pass filters respectively) (Dashkova et al., 2016). However, we did not use any fixators like glutaraldehyde because the count of photosynthetic cells can be underestimated (Troussellier et al., 1995).Moreover, the rainwaters were kept at 4°C in the dark (no growth and cell preservation), and we performed the flow cytometry shortly after the rain events (maximum of 48h), ensuring a realistic view of the environmental signature of the samples. The rainwaters, which could not be analyzed after 48h was discarded.

We report a different approach to quantifying the autofluorescence to measure specific pigment populations of photosynthetic cells rich in chlorophyll, phycocyanin, or phycoerythrin. An example is showed in the supplementary figure below. Indeed, we first select the photosynthetic cells population (total pigments) based on minimal chlorophyll fluorescence on the cytogram by selecting FL3 and SSC, opened on the preselected population without unwanted debris (Cells, plot A & B). Then, to distinguish the photosynthetic cells based on individual pigments autofluorescence, we create a new cytogram by selecting FL3 and FL2 channels taken from the "total pigments" population (plot C). This cytogram was divided into two: a "phycoerythrin rich population" and a not phycoerythrin-rich population. From the "NOT Phycoerythrin population," we create a new cytogram by selecting FL3 and FL4 channels divided into "chlorophyll" and "phycocyanin" populations (plot D). These selections allow us to separate the phycoerythrin population from chlorophyll or phycocyanin.

[Figure]

Suppl. figure: Flow cytometry based-method to quantify the autofluorescence from specific pigment from photosynthetic cells in the rain samples. Detection of all aerosols in the rain (A), the total pigments population based on minimal chlorophyll fluorescence (B), Phycoerythrin population (C), Chlorophyll, and Phycocyanin populations (C).

Figure 7A - The graphical scale for the water temperature does not cover the whole data spectrum - can the authors extend the x-axis from 6 to 20°C to include all data points?

The graph will be modified.

Line 470 - I disagree with the sentence. First because microalgae encompass both prokaryotic (cyanobacteria) and eukaryotic photosynthetic unicellular organisms. Second because previous studies have investigated the diversity of microalgae in wet depositions including rain. However, the methods used involved capture and growth, not rapid detection based on flow cytometry. The proposed sentence is therefore not proper, please rephrase.

Lines 473-475 - I also disagree with this sentence. One major problem with culturing is that not all organisms can grow in artificial media, therefore applying a selection pressure towards underestimating the environmental biodiversity. Another issue is that all isolated microalgae possess a biome (including bacteria). These bionts can be remove using diverse available methods. However, some microalgae need their bionts to survive. In any cases these should not impede microalgal detection using flow cytometry or micrococoscope-based techniques. Please reformulate the sentence.

To clarify lines 470-475, the sentence will be rephrased as below, including new references:

*"Moreover, the detection of photosynthetic cells by microscopy in rain samples is often measured on rain cultures after several days (Wiśniewska et al., 2022). However, this methodology seems inappropriate for estimating the natural environmental biodiversity because an unadapted artificial medium for the growth of all microorganisms could apply a selection pressure. Hence, we recommend using flow cytometry or microscopy-based techniques on fresh samples without fixators. Indeed, glutaraldehyde can lead to an underestimation of the count of photosynthetic cells and can alter the intensity of fluorescence according to the cell size* (Troussellier et al., 1995; Lepesteur et al., 1993). *»*

The English language and formulations need to be double checked by a native speaker, several mistakes are present in the text.

A native speaker will check the English mistakes.

References:

Dashkova, V., Segev, E., Malashenkov, D., Kolter, R., Vorobjev, I., and Barteneva, N. S.: Microalgal cytometric analysis in the presence of endogenous autofluorescent pigments, Algal Res., 19, 370–380, https://doi.org/10.1016/j.algal.2016.05.013, 2016.

Lepesteur, M., Martin, J. M., and Fleury, A.: A comparative study of different preservation methods for phytoplankton cell analysis by flow cytometry, Mar. Ecol. Prog. Ser., 93, 55–63, 1993.

Troussellier, M., Courties, C., and Zettelmaier, S.: Flow Cytometric Analysis of Coastal Lagoon Bacterioplankton and Picophytoplankton: Fixation and Storage Effects, Estuar. Coast. Shelf Sci., 40, 621–633, https://doi.org/10.1006/ecss.1995.0042, 1995.

---

## Author Comment (AC4)

**Comment on bg-2022-100**

**Referee #2**

Referee comment on "Interdisciplinary strategy to survey phytoplankton dynamics of a eutrophic lake under rain forcing: description of the instrumental set-up and first results" by Fanny Noirmain et al., Biogeosciences Discuss., https://doi.org/10.5194/bg-2022-100-RC2, 2022

We thank the referee for his suggestions made online and on the pdf. We respond in blue below to his comments (our answers to the major comments of the pdf have been added at the end of the text).

The study title "Interdisciplinary strategy to survey phytoplankton dynamics of a eutrophic lake under rain forcing description of the instrumental set-up and first results" by Noirmain et al. aims to define the fine scale effect of rain and the carried algal particles on a lake physiochemistry and phytoplankton community. They combine methodology from meteorological sciences analyzing cloud structure and origin and raindrop algal cytometry, with characterization of the water column properties and phytoplankton microscopy enumeration, in an innovative approach that aims to reconciliate recent findings of rain algal cell deposition (Dillon et al. 2020, Wisniewska et al. 2022, both cited thoroughly in the manuscript), with traditional works that explore the relationship between rain events and lake biogeochemistry, such as de Eyto et al. 2016 (DOI: 10.5268/IW-6.4.875). As far known, no efforts have been made to analyze the lake surface and the rain for both chemical composition and photosynthetic organisms.

The study has the potential to provide great insight into rain events effects in lake phytoplankton, short term surface stoichiometry and water column temperature changes. The evaluation of the photosynthetic organisms suspended in rain drops in comparison with the lake phytoplankton can provide great insight into the dispersal rate and mechanisms of the different phytoplankton organisms which is still a poorly understood subject. The inclusion of a phycocyanin channel (also with the chlorophyll channel provides a unique opportunity, together with the real time evaluation of the effects of rain on the physiochemistry of the lake, can provide a great insight on the effects of rain events on the phytoplankton community. Although the title of the article alludes to a presentation of experimental setup and first results, the listed objectives, rationale behind the analyses and discussion aim for a much definite style of work.

Unfortunately, the measurements provided seem disconnected. The water sampling times are too far away to assign any causality of the chemical and community changes to rain events and not wind induced or upstream inputs (which seems the case given the steady decrease of temperature in CR1). Mesocosm experiments could have helped isolate the rain effects from the basin and wind effects.

The suggestion to use a mesocosm is interesting and would help isolate rain to wind effects, but we do not have this material at our disposal and cannot perform this experiment. For this article, we focus on a case study of three rain events to illustrate a monitoring strategy of the atmosphere and lake. As we analyzed the rain impact on the lake stratification at a fine scale in real-time, we were able to report a causality between the rain intensity and the lake temperature. To better show the wind effect on the lake temperature, as suggested by the reviewer, we added a correlation between the water temperature and the wind during the rain events. So, during HIR, associated with low wind speed, we report an immediate

decrease in the diurnal surface water temperature. The correlation between the water temperature at the lake surface and the wind was positive but not significant (r=0.8, p-value = 0.3333). In contrast, the correlation with the rain intensity was significantly negative (r=-0.7, p-value=1.2.10$^{-06}$), suggesting the predominant water temperature decrease due to the rain amount at the surface.

On the other hand, during CR1 associated with a higher wind speed, the correlation between the water temperature and the rain intensity was lower (r=-0.24, p-value=1.3.10$^{-03}$), whereas those obtained with the wind speed was significantly negative (r=-0.63, p-value = 2.1.10$^{-03}$) until 3 m deep. Moreover, as the wind was higher during RP2, it could be argued that it accelerated the decrease of the lake temperature by mixing the water column during this period and weakening the lake stratification strength. This analysis and correlations with the wind could be added to the manuscript to show better the relative contribution of wind and rain in water temperature as the wind events.

Although we sampled the lake as soon as possible, we agree that a more frequent sampling would increase the understanding of the causality between biochemical rain and lake biochemical composition. Nevertheless, for this article, it was not the principal aim as we wanted to illustrate a strategy with high monitoring. Indeed, we present only three rain events as an example for the case study that does not allow us to connect precisely the rain and lake biochemical composition

The lake phytoplankton and the rain photosynthetic cells are measured with two different methods that are hard to reconcile. The rain cytometry does not seem to fix the cells with glutaraldehyde like Dillon et al. 2020, so the cells present in rain might be over or underestimated by growth or death in the rain collector chamber.

Overall, this work provides the first attempt to measure the rain effect on a lake on the fine scale. It can be improved by a bigger connection between the variables and timescales used to measure them and the questions set in the introduction. To properly answer said questions I suggest performing cytometry on the lake surface (and/or microscopy on rain samples), isolate the effects from rain and watershed inputs using mesocosm enclosures, and shortening the sampling time after the rain event. Although the main missing link is the disconnection of methods for phytoplankton analysis for the lake and for the rain, with comparative 2d cytograms of known cultures/species or lake samples, the two datasets can be made compatible. Regarding the manuscript structure, the introduction and the discussion need higher cohesiveness and streamlining, and I suggest that it undergoes severe rewriting. The results' figures could also benefit from trimming down to the ones specifically pertaining to the questions set.

The cytometry method used to detect the photosynthetic microorganisms in the rainwater was developed to cover the size of particles ranging from 0 to 30 μm. This methodology is well adapted for species in rainwater, as their size commonly found is under 30 μm in the atmosphere. On the contrary, the microscopy-based method is not well adapted due to the very low number of photosynthetic cells in rainwater. Moreover, in the literature, we found that some authors let the cells grow for 30 days before estimating the diversity, which leads to underestimating the diversity as all species cannot grow in an artificial medium (Wiśniewska et al., 2019). So we decided not to use this method.

On the contrary, the photosynthetic lake species can be composed of colonial or filamentous forms larger than 30 μm. Therefore, the flow cytometry is inappropriate and could lead to underestimating the

concentration of the lake's colonial or filamentous species. It is why we estimated the lake phytoplankton abundance by microscopy.

However, we also measured the phytoplankton by flow cytometry, using the same recorded parameters developed for the rainwater. An example is presented in the table below, showing a comparison between rain and lake samples measured by flow cytometry. The flow cytometry based-method detects small cyanobacteria (like *Synechococcus* and *Synechocystis*), which grow especially after RP2 (plot B, suppl. fig). On the contrary, the microscopy counts show that many cyanobacteria were present before RP1 and RP2 (plot C, suppl. fig), which were not detected by the flow cytometry based-method (plot B), corresponding to colonial cyanobacteria (*Microcystis*, *Merismopedia*...) (Fig 8B).

Although the sonication could help dissociate the colonial species, it could also damage cyanobacteria or flagellate cells, which will not be detected by flow cytometry. Moreover, chlorophyll a is not the best proxy to assess environmental diversity as the increase in chlorophyll a can be due to a higher proportion of large species (Felip and Catalan, 2000; Blottière et al., 2017).

Due to these technical issues and more clarity in this article, we select only the appropriate methods to detect the rain's photosynthetic cells by flow cytometry and the lake phytoplankton by microscopy.

**A**

Quantification of the photosynthetic cells from the pigment's fluorescence measured by flow cytometry on rain samples

[Figure]

**B**

Quantification of the photosynthetic cells from the pigment's fluorescence measured by flow cytometry on lake samples.

[Figure]

**C**

The microscopy could not be performed on rain samples due to the low number of photosynthetic cells

**D**

Quantification of the phytoplankton abundance by microscopy on lake samples (species were grouped by pigments type)

[Figure]

Suppl. Figure: Comparison of two methods, the flow cytometry and microscopy-based methods for the quantification of photosynthetic cells in lake and rain samples

The air mass analysis does not provide additional support to the questions set out in the introduction apart from a brief mention in the discussion about how CR rain was from the lower cloud system and not the higher one of marine origin. They discuss solar radiation and rain effects on the water column thermal structure, while avoiding wind effects, which is usually the dominant factor determining short term mixing changes (as clearly seen in the CR event).

In our view, it is essential to maintain the air mass analysis as it could explain one part of the dynamics of photosynthetic cells. However, with only the three rain events presented here, it is complicated to conclude about the origin of photosynthetic cells according to the air mass origin (from the sea or the continental, results are controversial on the topic), and more rain events analysis is necessary. Nevertheless, we aim to illustrate a strategy to investigate the potential link, and it seems crucial to keep such analyses in our article. Thus, we will improve the introduction to present this information more clearly and to improve the link between the questions set in the introduction and the discussion referring to the dynamic of photosynthetic cells in the rain. Likewise, we will add more bibliography citations related to the dynamic of these cells and will also add the following sentence in the introduction:

*"We illustrate a strategy to monitor the dynamic of photosynthetic cells in the rain by characterizing the cloud and rain physical properties, the meteorological variables, and the air mass origin."*

The discussion will also benefit in a cloud type-oriented organization, with sections for HIR and CR going each from the cloud source, the rain physiochemistry up to the lake chemistry changes and finally phytoplankton changes, instead of partitioning into their methodological counterparts. The text contains minor errors and will greatly benefit by proofreading by a native English speaker.

The cloud-type-oriented organization could be interesting. However, with only three rain events analyzed, we think it is premature at this stage as we did not perform a climatology study with a higher number of rain events. Nevertheless, this will be performed in fore coming studies.

With the current state of this work, it might be worth considering partitioning the work in two:

Analyzing the rain drops cytometry (and/or microscopy) and its relationship with the different sources and characteristics of the rain clouds, with additional 2d cytograms of representative samples of the lake or cultures to known in greater detail the composition of said cells. Optimally, cytometry of the lake surface before and after the event should be performed, but the timing should be precise to avoid changes due to algal migration.

It cannot be done that way because the culture will not give us more detail about the cells found in the rainwater (see the answer on the cytometry method above). Therefore, even if we used the culture of known species to adjust cytometry parameters, we have only information about the pigment types and the cell densities, which do not allow species identification. Moreover, 2d cytograms of known cultures/species or lake samples could not be compared with rain samples as species in culture do not represent all those found in the atmosphere (some of them are uncultivable). In any case, the flow cytometry could not give greater detail on the lake composition of said cells as we were limited in a size range to focus on the bioaerosols found in the rain and not those found in the lake.

Report on the lake phytoplankton and physiochemistry changes before and after different precipitation events but including a further discussion on wind effects and watershed inputs until a decoupling is achieved with mesocosm deployments.

Because we did not have a mesocosm at our disposal for comparisons, we cannot argue about the wind effects and watershed inputs.

Ions reported should be in the context of cyanobacterial biomass changes, i.e. NH4+, NO3-, and PO4-, and a point should be made of the viability of the rain droplets milieu as a growth media for the airborne cyanobacteria, while the other ones (that are used as fingerprints to identify the origin of the clouds) can be just mentioned in the text or in supplementary figures.

The atmospheric conditions are stressful for these species (UV, temperature variation, osmotic chock…). However, as mentioned in the literature, they could be viable, so maybe the nutrient composition favors their viability. We did not point out the viability of the rain droplets media because the rainwater is a temporary environment for the cells. Indeed, the photosynthetic cells can be scavenged in the cloud "long-range transport in the air mass" or washed out below the cloud from the air column. It is similar to ions. Hence, we did not know if their origin came from the cloud or the air and how long they stayed in this environment.

Specific comments included inline in the annotated pdf are enumerated here:

PDF comment for line 183: This 15 minute drought break seems very arbitrary and specifically to obtain 2 rain periods from what should be a single long one. Better support and justification is needed for this choices.

We agree that the two rain periods could be viewed as belonging to a larger rain event.

However, our goal is to study the possible high temporal links between rain and its origin and characteristics and its effect our lake water composition. Thus, it seems important to separate the different phases within a rain event as distinct periods, not only as a function of dynamics but also composition. In our example, it appears that the two separate periods actually show slightly different air mass origins, thus reinforcing our choice to select a drought period equivalent to the best interval to ensure a new bottle change between samples.

Line 202: Justify this normalization procedure.

The abundances of species (Fig8 A) were transformed into relative abundances to counter the heterogeneity in the number.

Fig 8A: Phytoplankton abundance should be avoided when talking about the whole community since it includes disparate taxa like Synechocystis (2 µm unicellular) and Microcystis (500+ µm colonies), biovolume/biomass should be reported instead.

The biovolume/biomass is particularly interesting in the food web study, which is not the case here. So, we prefer to keep the dynamics of phytoplankton in cellular concentration.

Line 545: concentration shouldn't be compared, given that the direct mass of rain is very small in comparison to the whole water column (accounting for the watershed is a different issue).

Line 548, we reported that the wet deposition did not influence the lake's chemical composition, certainly due to the small mass of rain compared to the lake (similar to the photosynthetic cells brought by wet deposition), very low in contrast to phytoplankton cells line 563).

Line 563: A discussion about phytoplankton dspersion and biogeography will greatly enhance this section.

We did not have the relevant information about species diversity in the rain samples. Thus, we cannot discuss the correlation between photosynthetic cells in the rain and the lake. Nevertheless, we will improve the introduction with citations describing the species encountered in the atmospheric compartments, especially in the rain.

Line 573: If they were "pulled down by mixing", then the charophytes and chlorophytes would have also decreased. Seems like these groups' buoyancy mechanisms created a downward migration. Lm and Lo codon have low surface/volume ratio on an individual basis, and not high as listed here. See: Reynolds 1994 (DOI: 10.1007/BF00007405)

We thank the referee for pointing out an inconsistency with the surface-to-volume ratio, which is low for Lm and Lo species. Therefore, we will rephrase the sentences as follow:

"*Microcystis*, *Merispomedia*, and *Coelomoron* belonging to Lm and Lo codons have a low tolerance to mixing. Therefore, their decrease after the rain events could be explained by a dilution caused by their transport through the entire mixed depth. However, when temporary stratification was back after the rain, a secondary thermocline nearer the surface could help these species to quickly regulate their vertical position by buoyancy mechanisms to benefit from optimal conditions (nutrient and water irradiance)."

In order to improve the discussion on wind effects, we will add sentences to describe the species increasing after the rains in link with the rain and wind events:

On the contrary, after the rain events, we reported a shift in species composition from colonial picocyanobacteria towards unicellular picocyanobacteria (*Synechococcus* and *Synechocystis*) and also the increase of diatoms (*Asterionella* and *Melosira*) and microalgae (*Elakatothrix*). Because *Synechococcus*, *Synechocystis*, and *Asterionella* have a high surface-to-volume ratio, they can quickly grow in the mixed layer depths if there is no light limitation. Indeed, Reynolds (1994) reported that optimal division rates of these species could lead to doubling their abundance per day when there is no light limitation. As there was no light-limitation during the lake campaign (the euphotic zone, 13 m deep, exceeds those of the thermocline, 7 m), it suggests that unicellular picocyanobacteria were able to develop to become dominant after the rain events.

On the other hand, we also reported the presence of larger unicellular algae, *Closterium*, present only after RP2, and the coenobial green algae *Elakatothrix*, which increased after the rain despite its low growth rate. Hence, as RP2 was characterized by stratiform rain events and high wind speed, we suggest that the wind increased diffusivity, allowing green algae to stay longer in the water column where no light-limitation was reported. Indeed, the mixing could favor larger and heavier species to stay suspended and their development, as has already been reported in the literature (Blottière et al., 2014).

References:

Blottière, L., Rossi, M., Madricardo, F., and Hulot, F. D.: Modeling the role of wind and warming on Microcystis aeruginosa blooms in shallow lakes with different trophic status, Theor. Ecol., 7, 35–52, https://doi.org/10.1007/s12080-013-0196-2, 2014.

Blottière, L., Jaffar-Bandjee, M., Jacquet, S., Millot, A., and Hulot, F. D.: Effects of mixing on the pelagic food web in shallow lakes, Freshw. Biol., 62, 161–177, https://doi.org/10.1111/fwb.12859, 2017.

Felip, M. and Catalan, J.: The relationship between phytoplankton biovolume and chlorophyll in a deep oligotrophic lake: decoupling in their spatial and temporal maxima, J. Plankton Res., 22, 91–106, https://doi.org/10.1093/plankt/22.1.91, 2000.

Reynolds, C. S.: The long, the short and the stalled: on the attributes of phytoplankton selected by physical mixing in lakes and rivers, Hydrobiologia, 289, 9–21, https://doi.org/10.1007/BF00007405, 1994.

Wiśniewska, K., Lewandowska, A. U., and Śliwińska-Wilczewska, S.: The importance of cyanobacteria and microalgae present in aerosols to human health and the environment – Review study, Environ. Int., 131, 104964, https://doi.org/10.1016/j.envint.2019.104964, 2019.

---

## Author Response (AR1)

**Author's Response:**

**Dear editor,**

Thank you for the detailed comments on our manuscript, which have helped to improve the manuscript's cohesiveness. We considered all the comments and questions carefully and have revised the manuscript accordingly. Please find below a point-by-point response with a list of the main changes (in blue).

**Reviewer #1**

"The manuscript by Noirmain et al entitled "Interdisciplinary strategy to survey phytoplankton dynamics of a eutrophic lake under rain forcing: description of the instrumental set-up and first results" is well written and easy to read. The described interdisciplinary strategy provides a very useful tool to decipher the impact of atmospheric processes on lake ecology.

An enormous work has been performed on the atmospheric sciences part but the authors only quickly discuss the ecological implications related to their study. The authors also need to consider further factors related the cyanobacteria biology and ecology. This could for instance apply to Lines 400-405 and Figures 8AB.A negative effect of rain on organism abundance was mentioned for few taxa. Effect can be seen on Fig. 8B on unicellular group forming cyanobacteria (e.g. Woronichinia, Microcystis, Merismopedia...). Whereas some unicellular individual cyanobacteria (e.g. Synechocystis, Synechococcus, Pseudoanabena...) seem to undergo a positive effect. Cell distribution factor is important because it impacts the growth dynamic and survival of the species. Moreover, microalgae can move and migrate through the water column. Almost the same concentration of cells was fund "before" and "after" RP events in the system (Fig. 8A). The increase concentration at a specific depth implies displacement of the phytoplanktonic community or organismal movement (RP2 towards 1.5 m depth, RP1 towards 3 m). Apparently deeper when the rainfall was more virulent (HIR). This is not described in the results nor correlated to Fig. 8B (diversity) but quickly mentioned in the discussion. Was the water column structure more stable (nutrient, temperature, light, water agitation...) at 1.5 m depth in RP2 and at 3 m in RP1?

All the cyanobacteria taxa retrieved have different ecological requirements. In the discussion Lm and Lo codons are rapidly mentioned. A principal component analysis can further help to characterize which cyanobacteria taxa was favored under certain rain events (e.g. peculiar nutrient signature).

As suggested by the reviewer, we carried out supplementary analysis by Multiple factor analysis (MFA) (Fig. 8 B-D) using the abiotic lake factors (water temperature and irradiance and amount of rain), phytoplankton taxa (described in Fig 8A), and the chemical composition of the lake to characterize which cyanobacteria taxa were more prolific after rain events. In the result part, we add sentences confirming the negative correlation between the amount of rain and *Microcystis, Merismopedia*, and *Coelomoron* abundances. We consider effect of the rain events on the picocyanobacteria (*Synechocystis*& *Synechococcus*), green algae (*Elakatothrix*), and diatoms (*Asterionella* and *Melosira*) to have been positive, as there was a correlation with the rain rate, as shown by multiple factor analysis (Fig 8 B-D) (lines 448-452). We also added sentences to discuss about the displacement of the phytoplankton, especially *Microcystis, Coelomoron*, and *Merismopedia*, which decreased systematically following the rain events. In contrast, *Synechococcus, Synechocystis, Elakatothrix* and *Asterionella* increased following the rain events and throughout the study on the lake, suggesting they are more affected by seasonal changes than any direct impact from rain (lines 619-630).

The water column does not seem more stable at 1.5 or 3 m after the rain events, as no statistical differences were found with depth (Fig 8D).

Moreover, the differences in phytoplankton concentration and taxa diversity between "before" and "after" calls for emission/deposition fluxes (e.g. Dillon et al 2020 doi:10.1128/AEM.01850-20, Mayol et al 2014 doi: 10.3389/fmicb.2014.00557), very briefly mentioned in discussion. However this is of ecological importance. Retrieved cyanobacteria species from the investigated lakes have been reported from atmospheric samples (e.g. Sharma et al 2007 DOI: 10.1111/j.1529-8817.2007.00373.x). Also microalgae can be emitted after local disturbances such as rain (Tesson et al 2016 doi:10.1128/AEM.03333-15, WiÅ>niewska et al 2019, 2022) from station Fig 1B2 and be redeposited in local system such as station Fig 1B1 or the lake, therefore it is possible that the retrieved peak of Cr1b with highest phycocyanin value could be partly indigenous.

To jointly study emission and deposition fluxes in Lake Aydat, we should sample airborne microalgae close to the lake's surface. Unfortunately, we did not measure the emission fluxes from the lake and so cannot directly link the phycocyanin detected in the rain (from one rain event, CR1) and the increase in picocyanobacteria abundance at the end of our lake sampling campaign. Nonetheless, we discuss deposition fluxes by comparing the level of photosynthetic cells collected in the rainwater with those in the lake, suggesting that a very low number of photosynthetic species was introduced into the lake by the wet deposition. We also discuss the deposition flux, which might impact the diversity in the lake by introducing different species collected by the rain (lines 590-602).

Moreover, it can be useful to run the analysis considering the ecology and behavior of these organisms to avoid a Simpson paradox."

We thank the reviewer for this suggestion and have added the multiple factor analysis in the revised version (Fig 8 B-D) and add also new correlations between the lake temperature and the rain rate and the wind speed (Fig.7). We also add new sentences in the discussion to consider the ecology and behavior of the organisms which have different abundances after rain events during the lake campaign (lines 609-630).

Figure 1 - Please add the geographical coordinates on Fig. 1A and the cardinal directions on both Fig. 1A and Fig. 1B.

Supplement Fig. 1 - The geographical coordinates are too small to be readable.

We created new maps with geographical coordinates for Fig. 1A & B and increased the size of the geographical coordinates in the supplementary figures illustrating the retro trajectory (Suppl. Fig.2).

Lines 111 and 113 - Inform about the material manufacturer and country between brackets.

Line 121 - Please add in the text the difference of elevation between the location where the instrumental setup was installed and the lake surface.

Lines 142 and 156 - Spell out the acronyms DSD and PCB.

The missing information was added in the materials and methods section (revised manuscript, modified text).

Lines 182-184 - How was the lake sampling performed? Which parameters were investigated? Do the authors refer to Lines 114-116 in situ measurements or were lake water- and/or phytoplankton samples collected? Please describe further.

**We added details about the lake sampling (lines 230-235).**

Line 198 - Why was a 10  $\mu$ m pore size used? In these filtrates, all microorganisms of a size inferior to 10  $\mu$ m would be present (including. bacteria, cyanobacteria and other <10  $\mu$ m eukaryotic microalgae), thus affecting the measurements. Moreover, -20°C storage was applied to the filtrates, conditions under which organismal cell damaging occur, releasing further nutrients in the water. Please explain further.

We thank the referee for pointing out an inconsistency with the pore size. We correct the sentences in the material and methods (lines 246 to 254 We filtrate the lake samples on a  $0.2\mu m$  nylon membrane before being kept at -20°C, avoiding the release of nutrients.

Line 200 - Which Lugol's iodine solution was used (acidic vs neutral)? What was the final concentration used? How and how long before investigation were the samples stored (temperature, darkness, time)? These are important information for cross studies comparison.

We added the missing information. (line 247).

Lines 221-236 - How does the flow cytometer detection method differ from existing protocols? (e.g. Haynes et al 2020, https://www.agilent.com/cs/library/applications/application-analysis-aquatic-plankton-novocyte-5994-2112en-agilent.pdf)

We added a new figure dealing with the flow cytometry method to the supplementary material (Suppl. Fig.1) to illustrate the cytograms created for the analysis and to show how the population of photosynthetic cells were isolated based on the presence of pigments (Chlorophyll, Phycocyanin, and Phycoerythrin).

Line 470 - I disagree with the sentence. First because microalgae encompass both prokaryotic (cyanobacteria) and eukaryotic photosynthetic unicellular organisms. Second because previous studies have investigated the diversity of microalgae in wet depositions including rain. However, the methods used involved capture and growth, not rapid detection based on flow cytometry. The proposed sentence is therefore not proper, please rephrase.

Lines 473-475 - I also disagree with this sentence. One major problem with culturing is that not all organisms can grow in artificial media, therefore applying a selection pressure towards underestimating the environmental biodiversity. Another issue is that all isolated microalgae possess a biome (including bacteria). These bionts can be remove using diverse available methods. However, some microalgae need their bionts to survive. In any cases these should not impede microalgal detection using flow cytometry or microcoscope-based techniques. Please reformulate the sentence.

We modify the sentences in the discussion.

The English language and formulations need to be double checked by a native speaker, several mistakes are present in the text.

The English language was checked by a native speaker.

**Reviewer #2**

"The study title "Interdisciplinary strategy to survey phytoplankton dynamics of a eutrophic lake under rain forcing description of the instrumental set-up and first results" by Noirmain et al. aims to define the fine scale effect of rain and the carried algal particles on a lake physiochemistry and phytoplankton community. They combine methodology from meteorological sciences analyzing cloud structure and origin and raindrop algal cytometry, with characterization of the water column properties and phytoplankton microscopy enumeration, in an innovative approach that aims to reconciliate recent findings of rain algal cell deposition (Dillon et al. 2020, Wisniewska et al. 2022, both cited thoroughly in the manuscript), with traditional works that explore the relationship between rain events and lake biogeochemistry, such as de Eyto et al. 2016 (DOI: 10.5268/IW-6.4.875). As far known, no efforts have been made to analyze the lake surface and the rain for both chemical composition and photosynthetic organisms.

The study has the potential to provide great insight into rain events effects in lake phytoplankton, short term surface stoichiometry and water column temperature changes. The evaluation of the photosynthetic organisms suspended in rain drops in comparison with the lake phytoplankton can provide great insight into the dispersal rate and mechanisms of the different phytoplankton organisms which is still a poorly understood subject. The inclusion of a phycocyanin channel (also with the chlorophyll channel provides a unique opportunity, together with the real time evaluation of the effects of rain on the physiochemistry of the lake, can provide a great insight on the effects of rain events on the phytoplankton community. Although the title of the article alludes to a presentation of experimental setup and first results, the listed objectives, rationale behind the analyses and discussion aim for a much definite style of work.

Unfortunately, the measurements provided seem disconnected. The water sampling times are too far away to assign any causality of the chemical and community changes to rain events and not wind induced or upstream inputs (which seems the case given the steady decrease of temperature in CR1). Mesocosm experiments could have helped isolate the rain effects from the basin and wind effects.

The suggestion to use a mesocosm is interesting and would help isolate rain from wind effects, but we do not have this material at our disposal and thus cannot perform this experiment nor discuss about the potential results.

To better illustrate the effect of wind on the lake temperature, as suggested by the reviewer, we added a correlation between water temperature and wind during the rain events (Fig. 7 B, D & F). We also included the Spearman correlation coefficients between the lake temperature and the rate of rainfall using the depths for the three rain events, HIR, CR1, and CR2. These relationships confirm that during HIR, the decrease in lake temperature at the surface was linked to the rain rate, and not the wind speed (no significant relationship). On the other hand, during CR1, when there was a higher wind speed, we found that the wind speed strongly impacted the lake temperature to a depth of 2.8 m. We added sentences in the results and in the discussion to describe these effects.

Although we sampled the lake as often as possible, more frequent sampling would improve our understanding of the causality between the biochemistry of the rain and biochemical composition of the lake. However, it was not our principal aim for this article, in which we wanted to illustrate a highly intensive monitoring strategy. Indeed, we present only three rain events as examples for our case study. However, two days after HIR, our results seem to show no direct link between the chemical composition of the rain and the lake's inorganic ion concentration, as they have an opposite trend after the rain events (lines 578-588). Furthermore, the phytoplankton dynamic shows a similar pattern in its assemblage, linked to the rainfall rate rather than the composition of the rain events.

The lake phytoplankton and the rain photosynthetic cells are measured with two different methods that are hard to reconciliate. The rain cytometry does not seem to fix the cells with glutaraldehyde like Dillon et

**al. 2020, so the cells present in rain might be over or underestimated by growth or death in the rain collector chamber.**

The cytometry method used to detect the photosynthetic microorganisms in rainwater was developed to cover particles ranging from 0 to 30  $\mu$ m in size. This methodology is well suited to species in rainwater, as they are generally under 30  $\mu$ m in the atmosphere. However, the microscopy-based method is not well adapted due to the very low number of photosynthetic cells in rainwater. Hence our choice not to use this method.

On the other hand, the photosynthetic species in the lake can be composed of colonial or filamentous forms larger than 30  $\mu$ m. Therefore, the flow cytometry is inappropriate and could lead to an underestimation of the concentration of colonial or filamentous species in the lake. This is why we chose to estimate the lake phytoplankton abundance by microscopy. Due to these technical issues, which we have described in more detail now in the article, we selected only the appropriate methods to detect the photosynthetic cells in rain by flow cytometry and the lake phytoplankton by microscopy.

We did not use any fixators like glutaraldehyde as it can lead to an underestimation of the count of photosynthetic cells and can alter the intensity of fluorescence according to the cell size (Troussellier et al., 1995; Lepesteur et al., 1993). Moreover, the rainwaters were kept at 4°C in the dark (no growth and cell preservation), and we performed the flow cytometry shortly after the rain events (maximum of 48h), ensuring a realistic view of the environmental signature of the samples. We precise the stored condition lines 242.

Overall, this work provides the first attempt to measure the rain effect on a lake on the fine scale. It can be improved by a bigger connection between the variables and timescales used to measure them and the questions set in the introduction. To properly answer said questions I suggest performing cytometry on the lake surface (and/or microscopy on rain samples), isolate the effects from rain and watershed inputs using mesocosm enclosures, and shortening the sampling time after the rain event. Although the main missing link is the disconnection of methods for phytoplankton analysis for the lake and for the rain, with comparative 2d cytograms of known cultures/species or lake samples, the two datasets can be made compatible. Regarding the manuscript structure, the introduction and the discussion need higher cohesiveness and streamlining, and I suggest that it undergoes severe rewriting. The results' figures could also benefit from trimming down to the ones specifically pertaining to the questions set.

We characterized the cloud-types as stratiform or convective at the beginning of the introduction, based on their physical characteristics and forecasting probabilities (lines 50-59). We rephrased the next paragraph that details the rain's impact on abiotic and biotic changes to the lake (lines 58-86). We added a new paragraph to the introduction relating to the dynamics of photosynthetic cells in the rain, their scavenging by the rain, and the potential consequences of their scavenging on a freshwater ecosystem (lines 88-98). We rephrased the main questions involved in the manuscript to correspond to our results and discussion (lines 100-117).

It was not possible to use the same methodology to estimate the concentrations of the photosynthetic cells in the rain and the phytoplankton. We cannot perform cytometry on lake samples due to the cell size larger than 3  $\mu$ m, and we cannot perform microscopy on rain samples due to the low number of cells (even after concentration of the rainwater by ultrafast filtration). An example of the incompatibility of the flow

cytometry for the phytoplankton present in the lake was presented in the interactive discussion of the journal, with results on natural samples showing an underestimation of the concentration of the lake's colonial or filamentous species by flow cytometry. Due to these technical issues, we estimated the lake phytoplankton abundance by microscopy and the photosynthetic cells in the rain by flow cytometry.

The air mass analysis does not provide additional support to the questions set out in the introduction apart from a brief mention in the discussion about how CR rain was from the lower cloud system and not the higher one of marine origin. They discuss solar radiation and rain effects on the water column thermal structure, while avoiding wind effects, which is usually the dominant factor determining short term mixing changes (as clearly seen in the CR event).

In our view, it is essential to maintain the air mass analysis as it could explain part of the dynamics of photosynthetic cells. We developed the discussion about the link between the photosynthetic cells and the origin of the air mass (lines 520-529). We also discuss the impact of wind on the lake temperature during the period of the lake campaign (lines 563-573).

The discussion will also benefit in a cloud type-oriented organization, with sections for HIR and CR going each from the cloud source, the rain physiochemistry up to the lake chemistry changes and finally phytoplankton changes, instead of partitioning into their methodological counterparts. The text contains minor errors and will greatly benefit by proofreading by a native English speaker.

The cloud-type oriented organization could be interesting. However, since we only analyzed three rain events, we think it would be premature at this stage. This will be carried out in future studies involving a higher number of rain events.

With the current state of this work, it might be worth considering partitioning the work in two: Analyzing the rain drops cytometry (and/or microscopy) and its relationship with the different sources and characteristics of the rain clouds, with additional 2d cytograms of representative samples of the lake or cultures to known in greater detail the composition of said cells. Optimally, cytometry of the lake surface before and after the event should be performed, but the timing should be precise to avoid changes due to algal migration.

This cannot be carried out because the culture will not provide us with more details about the cells found in the rainwater. Therefore, even if we used the cultures of known species to adjust cytometry parameters, we would only have information about the pigment types and the cell densities, which do not allow species identification. Moreover, 2d cytograms of known cultures/species or lake samples could not be compared with rain samples as species in a culture do not include all of those found in the atmosphere (some of them are uncultivatable). In any case, the flow cytometry could not provide any more details about the lake composition of these cells as we were limited in size range to focus on the bioaerosols found in the rain and not those found in the lake.

Ions reported should be in the context of cyanobacterial biomass changes, i.e. NH4+, NO3-, and PO4-, and a point should be made of the viability of the rain droplets milieu as a growth media for the airborne cyanobacteria, while the other ones (that are used as fingerprints to identify the origin of the clouds) can be just mentioned in the text or in supplementary figures

We added a paragraph mentioning the viability of the photosynthetic cells in the rain (lines 529-535) and detailed the hostile conditions facing these species in the atmosphere (UV, temperature variation, osmotic

shock, etc.). We also added the plot showing the ion concentrations in the rainwater in supplementary material (Suppl. Fig.5).

PDF comment for line 183: This 15 minute drought break seems very arbitrary and specifically to obtain 2 rain periods from what should be a single long one. Better support and justification is needed for this choices.

Regarding the rain sampling procedure, we justified the 15-minute drought break in the rain sampling methodology by referencing a previous study showing a significant variability in the rain's chemical composition based on the air mass variability (lines 171-174). The authors of this study noted rapid changes in the atmospheric concentrations due to aerosol scavenging over a 10-minute period.

Fig 8A: Phytoplankton abundance should be avoided when talking about the whole community since it includes disparate taxa like Synechocystis (2  $\mu$ m unicellular) and Microcystis (500+  $\mu$ m colonies), biovolume/biomass should be reported instead.

As recommended by the reviewer, we selected only those results that were relevant to the questions laid out in the introduction. Therefore, we discarded the plot showing phytoplankton abundance.

Line 545: concentration shouldn't be compared, given that the direct mass of rain is very small in comparison to the whole water column (accounting for the watershed is a different issue).

Regarding our results, we reported that the wet deposition did not influence the lake's chemical composition, certainly due to the small volume of rain compared to the lake one (lines 578-588). However, as the literature suggested that wet depositions from the atmosphere can affect the dissolved organic carbon and nitrogen concentrations at the lake surface (line 61), we think it is interesting to make the comparison between the chemical composition of the rain and those of the lake, especially after HIR, where the lake was sampled after 2 days following the event. We also added the need to confirm these results by increasing the lake sampling at different times after the rain events (line 588).

Line 563: A discussion about phytoplankton dspersion and biogeography will greately enhance this section. We did not have the relevant information about species diversity in the rain samples. Thus, we cannot discuss the correlation between photosynthetic cells in the rain and the lake.

Line 573: If they were "pulled down by mixing", then the charophytes and chlorophytes would have also decreased. Seems like these groups' buoyancy mechanisms created a downward migration. Lm and Lo codon have low surface/volume ratio on an individual basis, and not high as listed here. See: Reynolds 1994 (DOI: 10.1007/BF00007405)

We modified the sentence to develop the hypothesis about the downward migration of some species after the rain events (lines 608-617). We discussed the species' tolerance to mixing and added results from the literature to explain our findings. In addition, we discussed the genera that increased after the rain events.

Best regards,

Fanny Noirmain

---

## Author Response (AR2)

**Author's Response:**

**Dear editor,**

Thank you again for your attentive lecture and the assiduous comments on our manuscript. We carefully considered all the comments and questions and revised the manuscript accordingly. Please find below a point-by-point response with a list of the main changes (in blue).

**Reviewer #1**

**Comments:**

Figure 5 - Can the authors please double check the Na+ concentration distribution for the three events. It seems weird to see the highest Na+ concentration in an air mass originating from brackish water and land (CR2, Fig 4 A-D) and a low Na+ concentration coming from open sea (marine water) and land (HIR, Fig 3A).

We checked the data and we confirmed the Na+ concentration values for the three events. The high value of Na+ measured during CR2 was associated with a high value of Cl-. The ratio Cl-/Na+ during CR2 (1.16) was closer to the seawater ratio (1.1), confirming a marine influence from the Mediterranean and Black Seas at low altitudes (Suppl. Fig 2 F).

During HIR, the seawater ratio was higher than the seawater ratio, reaching 1.75, confirming the open sea water influence and the presence of anthropogenic chloride wet deposition (Salve et al., 2008; Keresztesi et al., 2019). The difference in the ion concentrations could be a consequence of possible chemical reactions and/or loss of chemical species during the passage from sea to land (Möller, 1990). Finally, if previous wet deposition occurred before the rain events recorded at Aydat, an underestimation of the sodium concentration in the rain could be reported during HIR for example.

Line 354 – Please remind the reader how many samples were collected in CR1. Here the detection of phycocyanin was in 2 (out of how many samples?), or perhaps call for figure 6.

Moreover, to facilitate the read for this paragraph, knowing that the paper is directed to a broad audience, I believe it is important that the authors mention which kind of microalgae they are expected to detect with phycocyanin signal (which microalgae possess phycocyanin pigment) and which one the phycoerythrin.

The CR1 event was composed of 3 intra-events named CR1a, CR1b and CR1c (Fig. 6). The figure 6 is mentioned in the text line 358.

Line 292-293: We completed the material and method by mentioning the group of phytoplankton that are detected in flow cytometry for each gate. Hence, the "chlorophyll" gate allows only species without the phycocyanin or phycoerythrin pigments, such as Chlorophyceae, Diatoms, and Chrysophyceae. The phycocyanin gate allows the detection of blue-green cyanobacteria whose major pigment is C-Phycocyanin. Finally, the "phycoerythrin" gate detects the pink to yellow cyanobacteria and the Cryptophyceae (Read et al., 2014). However, some pollen may also be detected in the "phycoerythrin" gate.

**Lines 327-328 and 497 - These water systems are brackish waters.**

We mentioned in the text brackish water instead of marine source to discriminate from open sea.

Line 506 - The terminology is not proper! as mentioned in ulterior review report of this manuscript, the term microalgae encompasses both eukaryotic (i.e. bacillariophyta, charophyte, chlorophyll, miozoa) and prokaryotic (i.e. cyanobacteria) organisms. A possibility is to write "information about eukaryotic microalgae and cyanobacteria in rainwater...".

**We thank the reviewer for pointing this imprecision which has been corrected by adding the term eukaryotic before microalgae (line 509).**

Lines 590-604 - This is a rather simplistic explanation that works within the time frame of this experiment. The authors observed the phytoplankton dynamic over max 2 weeks. However, the authors should not neglect the biology and ecology of the species. Many studies (including limnic studies) have been performed on species introduction and species invasion. Long term analyses would be more suited to discuss about the repercussion on lake ecology. Note that the thematic is better expressed in the conclusion section.

We added sentences in discussion (lines 612-617) and conclusion (lines 671-674) to discuss about the limitation of the method.

**Minor comments:**

Line 73 - There are a couple of issues in this sentence. 1) Either some words are missing or "and could" needs to be removed. 2) Precise an increase of what; abundance or perhaps diversity?

**We removed "and could" and added the term abundance**

Line 90 - It seems that words are missing in this sentence: "However, their being washed out". Perhaps "their capacity at"?

**We corrected the sentence**

Figure 1B - The figure contains two diagonals of unexpected green pixels.

The green pixels are not present on the picture into the PDF, a new upload in a different format should correct this.

Figure 1C - To increase readability, please indicate that the elevations between brackets are above the sea level.

On the figure 1C, we added the cloud attitude level from the sea level instead to the ground level.

Line 347 - Do the authors use the usual 5% significance level? In the case of p=0.066 for the sodium ion concentration, the difference is only marginally significant. Please check again this value or adjust the text.

The significance level used was 5 % but we drop a 0 at the p-value which is 0.0066 (\*\*).

Lines 351-352 / 381-386 - The unit is usually given in cells.L-1

We added an "s" at the unit.

Line 453 - How was the positive relationship assessed, statistically or from visual observations from graphs?

The positive relationship was assessed statistically by Multiple factor analysis, which are representing by the correlation circle on the figure 8 C.

**Line 650 – "highlight" instead of "highlighting"**

We corrected the sentence

Best regards,

Fanny Noirmain

**References:**

Keresztesi, Á., Birsan, M.-V., Nita, I.-A., Bodor, Z., and Szép, R.: Assessing the neutralisation, wet deposition and source contributions of the precipitation chemistry over Europe during 2000–2017, Environ. Sci. Eur., 31, 50, https://doi.org/10.1186/s12302-019-0234-9, 2019.

Möller, D.: The Na/CL ratio in rainwater and the seasalt chloride cycle, Tellus B, 42, 254–262, https://doi.org/10.1034/j.1600-0889.1990.t01-1-00004.x, 1990.

Read, D. S., Bowes, M. J., Newbold, L. K., and Whiteley, A. S.: Weekly flow cytometric analysis of riverine phytoplankton to determine seasonal bloom dynamics, Environ. Sci. Process. Impacts, 16, 594–603, https://doi.org/10.1039/C3EM00657C, 2014.

Salve, P. R., Maurya, A., Wate, S. R., and Devotta, S.: Chemical composition of major ions in rainwater, Bull. Environ. Contam. Toxicol., 80, 242–246, https://doi.org/10.1007/s00128-007-9353-x, 2008.